



# EUREC⁴A's Maria S. Merian ship-based cloud and micro rain radar observations of clouds and precipitation.

Claudia Acquistapace[1], Richard Coulter[2], Susanne Crewell[1], Albert Garcia-Benadi[3,4], Rosa Gierens[1], Giacomo Labbri[6], Alexander Myagkov[5], Nils Risse[1], and Jan H. Schween[1]

[1]Institute for Geophysics and Meteorology, University of Cologne, Pohligstrasse 3, 50969, Koeln, Germany (DE)
[2]Argonne National Laboratory, 9700 S Cass Ave, Lemont, IL 60439, United States (US)
[3]Department Applied Physics—Meteorology, Universitat de Barcelona, Barcelona, 08028 (ES)
[4]Universitat Politècnica de Catalunya, Vilanova i la Geltrú, 08800, Spain
[5]RPG Radiometer Physics GmbH, Werner-von-Siemens-Straße 4, 53340 Meckenheim, Germany (DE)
[6]Universita' di Bologna, Via Zamboni 33, 40126 Bologna, Italy (IT)

**Correspondence:** Claudia Acquistapace (cacquist@meteo.uni-koeln.de)

**Abstract.** As part of the EUREC⁴A field campaign, the research vessel Maria S. Merian probed an oceanic region between 6° N and 13.8° N and 51° W to 60° W for approximately 32 days. Trade wind cumulus clouds were sampled in the trade-wind alley region east of Barbados as well as in the transition region between the trades and the intertropical convergence zone, where the ship crossed some mesoscale oceanic eddies. We collected continuous observations of cloud and precipitation profiles at

unprecedented vertical resolution (7-10 m in the first 3000 m) and high temporal resolution (1-3 s) using a W-band radar and micro-rain radar (MRR-*PRO*), installed on an active stabilization platform to reduce the impact of ship motions on the observations. The paper describes the ship motion correction algorithm applied to the Doppler observations to extract corrected hydrometeors vertical velocities and the algorithm created to filter interference patterns in the MRR-*PRO* observations. Radar reflectivity, mean Doppler velocity, spectral width and skewness for W-band and attenuated reflectivity, mean Doppler velocity

and rain rate for MRR-*PRO* are shown for a case study to demonstrate the potential of the high resolution adopted. As non-standard analysis, we also retrieved and provided liquid water path (LWP) from the 89 GHz passive channel available on the W-band radar system. All datasets and hourly and daily quicklooks are publically available. Data can be accessed and basic variables can be plotted online via the intake catalog of the online book "How to EUREC⁴A".

## 1   Introduction

Clouds and precipitation in the tropics are crucial for radiative budget and are responsible for climate prediction uncertainties (Bony and Dufresne, 2005). From 19 January 2020 to 19 February 2020, the "EUREC⁴A: A Field Campaign to Elucidate the Couplings Between Clouds, Convection and Circulation" campaign (Bony et al., 2017) took place in the Atlantic waters southeast of Barbados to test hypotheses on trade wind cumuli cloud feedbacks. Stevens et al. (2021) describe how the campaign's initial scope greatly expanded towards additional research questions, extending the campaign area and the number of scientific

platforms involved. To understand the factors influencing rain formation, study the evolution of mesoscale oceanic eddies and their impact on air-sea interactions, and produce a dataset that can stand as a benchmark for future model evaluations





and satellite retrievals became complementary goals of the enlarged campaign. The project EUREC[4]A-OA was granted two research vessels (RVs) in the Atlantic sea south-east of Barbados to monitor the oceanic processes induced by large-scale oceanic eddies.

The RV Maria Sybilla Merian (MS Merian) was deployed in the southern part of the EUREC[4]A domain to investigate how mesoscale oceanic eddies impact oceanic circulation and their role in cloud and precipitation formation. The collaboration with the ARM Mobile Facility 2 (https://www.arm.gov/capabilities/observatories/amf) equipped the RV with a comprehensive suite of remote sensing instrumentation that could track each stage of the precipitation life cycle. A micro rain radar (MRR-*PRO*) and a cloud radar (W-band) were installed on a stabilization platform: while the W-band radar is sensitive to a wide range of
atmospheric scatterers from tiny cloud drops to raindrops, the MRR-*PRO* can adequately describe the sub-cloud layer's rain evolution. The 89 GHz passive channel available in the W-band radar system allowed to characterize the columnar amounts of liquid water. The collaboration with ARM and the use of their stabilization unit allowed the compensation for ship motion and for the first time, made possible to obtain essential Doppler observations at unprecedented spatial and time resolution of the entire precipitation life cycle.

This radar suite represents one of the most advanced remote-sensing setup for measuring trade wind precipitation in and below the cloud. Ground-based cloud radar remote sensing has been used for long time to monitor the vertical structure of clouds and precipitation (Bretherton et al. (2010), Lamer et al. (2015), Leon et al. (2008), Kollias et al. (2007), ), as well as on ships (Zhou et al., 2015). In recent years, the potential of new observables like the Doppler spectra's skewness to detect precipitation forming in the cloud (Kollias et al. (2011b), Kollias et al. (2011a), Luke and Kollias (2013), Acquistapace
(2017)) was demonstrated for fixed ground-based sites. However, ship-borne cloud radar Doppler measurements have not been exploited yet. A first analysis of the unique dataset of trade wind cumulus clouds and precipitation collected with the MRR-*PRO* and the W-band radar on MS Merian is presented. Considering typical sea wave periods of 9 s, to obtain Doppler observations at sea, integration times have to be chosen shorter than 1 s (Chris Fairall, personal communication). In the paper, we document how specific choices on the integration times of the instruments were taken, describing the measurement sampling strategy
regarding spatial and temporal resolution.

The synergistic usage of the dataset collected on the RV will be crucial for tackling precipitation life cycle detection using a multiscale approach based on the additional measurements onboard: a water vapor Raman lidar and a wind lidar from the University of Hohenheim, a cloud kite from the Max Planck Institute for Dynamics and Self-Organization of Göttingen (http://www.lfpn.ds.mpg.de/MCO/ck.html), that is a 250 $m^3$ balloon able to fly up to 2500 m for in-situ observations of cloud
and raindrop size distributions, 3d wind profiles, and eddy dissipation rates. When combining the W-band radar and the co-located in situ observations from the cloud kite, detailed description of the precipitation process and unique reference data for high resolutions model runs become available. The high vertical (7-10 m) and temporal (1-3 s) resolution adopted by all the active remote sensing instrumentation below 2500 m will constitute an essential benchmark for future satellite missions like EarthCare (Illingworth et al., 2015), providing a detailed description of the atmospheric layer closer to the surface that is and
will be the most critical region to detect from satellite (Lamer et al., 2020). The 1-month precipitation data collected during the campaign also represents a vital evaluation dataset for Global Precipitation Measurement mission (GPM) performance at sea





in the subtropics for shallow convection precipitation (Hou et al., 2014). The stabilization platform worked for approximately 65% of the time, while for 35% of the time it did not and we considered ship motion corrections for both situations. The track followed by the RV MS Merian allows to characterize the latitudinal dependency on the cloud fields when moving from the
subtropics towards the inter-tropical convergence zone and understand the impact of the sea surface temperature heterogeneities on the boundary layer (Laxenaire et al., 2018). Active remote sensing instruments on a stabilization platform were installed on the RV MS Merian for the first time. The obtained dataset can be a reference dataset for further analysis like process studies, model evaluations and comparison of satellite retrievals. We collected some lessons learned during the EUREC[4]A campaign with the hope of encouraging and facilitating future deployments of active remote sensing instruments on ships, given the
strategic importance that such data might have.

The paper is organized as follows. Section 2 describes the experimental setup and the instrument characteristics. Section 3 provides details on the data processing and on the removal of the interference pattern from the data, assessing the impact of the ship motion correction algorithm. Section 4 describes a case study of trade wind cumulus clouds and precipitation. We describe how to access data and processing scripts in section 5, while Session 6 briefly collects the lessons learned and 7 summarizes
the work.

## 2 Experimental setup

We positioned the radar equipment on the RV's top deck at around 20 m above sea level, as far as possible from the influence of sea spray (Figure 1). The W-band radar and the MRR-*PRO* were mounted on the stabilization platform using two metal bars. To limit vibrations, we installed rigid support between the MRR-*PRO*'s pole and the W-band radar. We calibrated the receiver
of the W-band radar after installing it in the position shown in Figure 1. The receiver calibration is done using two calibration targets with two distinct brightness temperatures as described in Küchler et al. (2017). The MRR-*PRO* comes with a factory calibration. At installation, we synchronized the internal clocks of the computers controlling the radar equipment with the ship navigation system clock. Despite this effort, the time stamp synchronization suffered from a drift of the clocks with respect to the Global Positioning System (GPS) time of the ships inertial system that we had to consider in the correction of the data for
ship motions. From 19 January 2020 to 19 February 2020, the RV MS Merian sailed over a vast oceanic region spanning from 6°N to 13.8°N and from 51°W to 60°W (see Figure 2). We launched 118 radiosondes and collected 38 descents to provide temperature (T), pressure (P) and humidity (q) profiles during the whole campaign (Stephan et al., 2021). This database was used to build a retrieval for integrated water vapor from the single passive 89 GHz channel available on the W-band radar. In the following, we will first describe the two radars followed by the stabilization platform and we will describe the Motion
reference unit (MRU) and the ship reference system.

### 2.1 W-band radar

The W-band radar is a frequency modulated continuous-wave (FMCW) 94 GHz dual polarization radar equipped with a radiometric channel at 89 GHz and is manufactured by Radiometer Physics GmbH (RPG), Germany. The small diameter of its

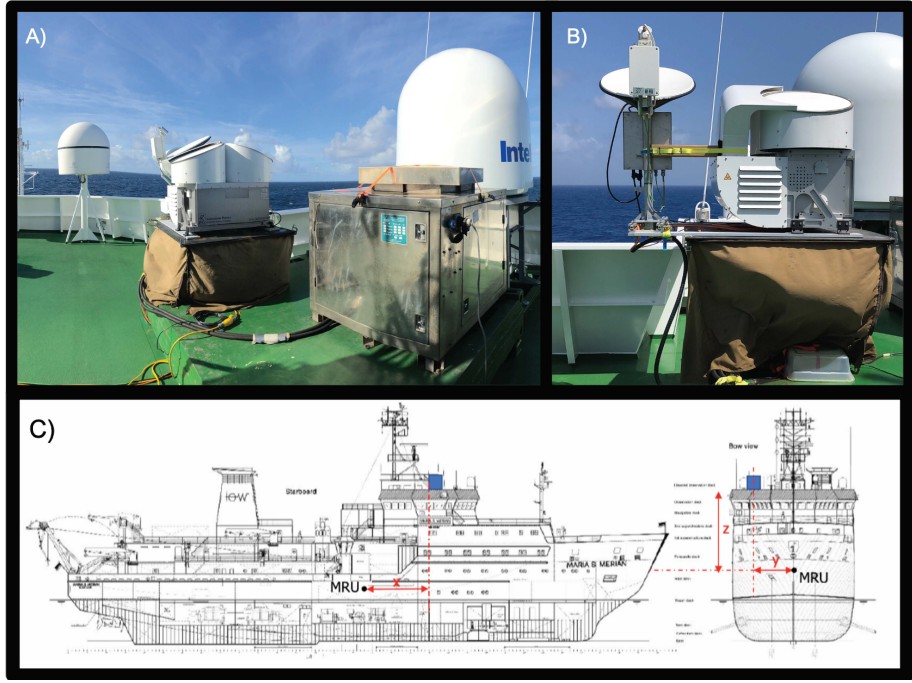

**Figure 1.** Instrument deployment on the RV MS Merian. A) View of the top deck: the hydraulic unit is visible on the right in the metal box, connected via multiple cables to the stabilization platform. B) The MRR-*PRO* (left) and the W-band radar (right) fixed using two metal bars on the stabilization platform. C) Position of the instrument deployment with respect to the Motion Reference Unit (MRU) on the MS Merian: side view (left) and front view (right).

antennas (0.5 m), one to transmit and one to receive, and its compactness (Table 1) make it a well suited instrument to be de-
ployed in complex environments. Küchler et al. (2017) provided an extended description of the radar performance, hardware, calibration and signal processing procedures. To protect the hydrophobic radome from hydrometeors, the radar is equipped with a blower for both antennas. The blower is able to produce a thin airflow with up to 20 ms$^{-1}$ over the antenna radomes (Küchler et al., 2017). Users can set different range resolutions at different altitudes by providing the necessary parameters to the so called "chirp table", i.e. a table storing all the frequency modulation settings. Table 2 shows the chirp table definition
adopted for this measurement campaign. We defined the chirp table to have a high vertical resolution below the inversion layer to focus on shallow cumulus clouds (Table 2). This choice resulted in reaching a maximum detectable range of 10000 m to focus on high vertical resolution of the boundary layer clouds and the inability to measure high cirrus clouds. The range resolution from the sea-level to 1233 m was 7.5 m, while it was 9.2 m between 1233 m and 3000 m. Between 3000 m and 10000 m the range resolution was 34.1 m. We chose integration times of 0.846 s for heights smaller than 1233 m, 0.786 s between 1233 and 3000 m, and 1.124 s between 3000 m and 10000 m to make the ship motion correction effective. The total sampling time
required to measure a full profile resulted in around 3 s.

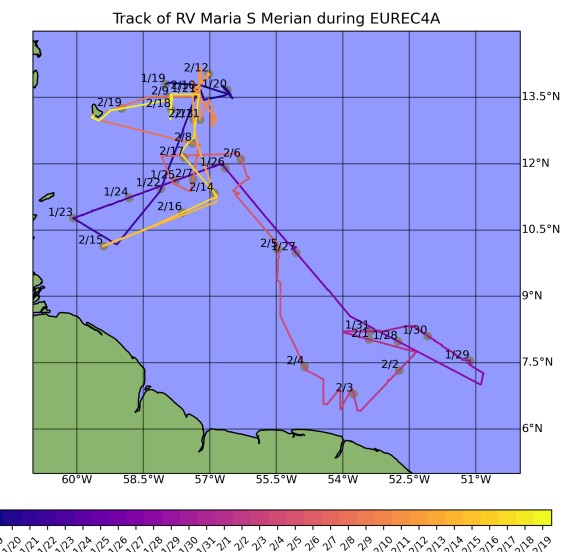

**Figure 2.** Ship track of the RV MS Merian during the EUREC⁴A campaign, from 19 January to 19 february 2020.

The embedded passive channel operates at 89 GHz with a bandwidth of 2 GHz and measures the calibrated brightness temperature (TB). In the W-band, atmospheric gases are relatively transparent. The absorption coefficient of atmospheric gases in the lower troposphere is of the order of 1 dB/km (Ulaby et al., 1981). In contrast, cloud liquid water produces a strong

attenuation ($\approx 1$ dB km$^{-1}$g$^{-1}$m$^3$, (Ulaby et al., 1981)) in this frequency band. Since the passive measurements are sensitive to the presence of liquid water, the TB measured at 89 GHz can be used in a retrieval of liquid water path (LWP) (Küchler et al., 2017). The cloud radar continuously runs a statistical retrieval developed by the radar manufacturer. The retrieval is based on an artificial neural network (ANN), which approximates values of LWP for a given set of the observed TBs, surface temperature, relative humidity, pressure, and day of the year. For the ANN training, a dataset of atmospheric profiles was used. Since the

ship was moving around Barbados during the campaign, data from several surrounding stations were combined in the dataset. The dataset consisted of 3 radiosonde stations and one ERA-Interim reanalysis column (Dee et al., 2011). For more details in the ANN and the dataset used for it, please refer to the Appendix A and Table A1.

We obtained integrated water vapor (IWV) estimates by applying a single-channel retrieval in clear sky cases, defined as profiles where all W-band radar reflectivity values are smaller than -50 dBz. The retrieval is based on the quadratic regression

between the 89 GHz brightness temperatures and the IWV estimated with the radiosoundings. We selected all radiosoundings with relative humidity smaller than 97% in the entire profile launched when no cloud base was detected by the wind lidar on-board MS Merian. The IWV results from integrating the profile of specific humidity over height, and is associated with the mean 89 GHz brightness temperature calculated over 1 minute after the radiosonde launch.





The W-band radar data collected during the EUREC[4]A campaign have been post-processed using a software package, that
includes processing and de-aliasing of compressed and polarized spectra. The code is an update and a subsequent restructuring
of the first program version provided by Küchler et al. (2017) and it is available at https://github.com/igmk/w-radar/tree/new_
output_structure. No attenuation correction has been applied to the data yet. The post-processing routine produces as output
a technical data file including all radar specific variables, and a physical data file, available in two versions. One version
(compact) includes:

- radar moments (equivalent reflectivity factor, mean Doppler velocity negative towards the ground, Doppler spectral
  width, Doppler spectrum skewness, Doppler spectrum kurtosis)

- coordinates (time, height)

- integrated variables (liquid water path, brightness temperature at 89 GHz)

- surface variables collected by the meteo station attached to the radar (wind speed and direction, pressure, temperature,
  rainfall rate and humidity),

- general parameters (filecode/version number, compression flag)

The other version (complete radar data) includes all the previous variables plus additional radar variables like the Doppler
spectrum, the bin mean noise power and the sensitivity limit. In addition to the standard processing, we derived and added the
mean Doppler velocity field corrected for ship motions to the variables listed above in both versions of the files. The compact
version has been enhanced with Climate and Forecast (CF) conventions (https://cfconventions.org/) to allow online plotting
using the EUREC[4]A book (https://howto.eurec4a.eu/intro.html). Section 3.1 and 3.2 describe the post-processing applied to
the data and section 5 explains the available data products.

## 2.2 Micro rain radar

The Micro Rain Radar (MRR-*PRO*) deployed on the RV MS Merian is a vertically pointing frequency modulated continuous-
wave (FMCW) Doppler radar operating at 24.23 GHz, produced by the Meteorologische Messtechnik GmbH (Metek) (Peters
et al., 2002) and owned by University of Leipzig. The instrument deployed was the latest version of the MRR, the so-called
MRR-*PRO*, with antenna diameter of 0.6 m (Figure 1). Table 1 contains the main technical characteristics of the MRR-*PRO*
and the specific settings adopted during the campaign. During the course, we observed interference of the instrument with the
stabilization platform device. For this reason, we postprocessed the data independently instead of relying on the postprocessing
of the manufacturer. Details on the ship motion correction algorithm and the interference filter are provided in section 3.2
and 3.4, respectively. The data are organized in daily files and the variables provided after the processing chain described in
Figure 3 are reflectivity considering only liquid drops, equivalent reflectivity non-attenuated, equivalent reflectivity attenuated,
hydrometeor fall speed, spectral width, skewness and kurtosis of the Doppler spectra, liquid water content, rainfall rate, rain
drop size distribution, raindrop diameter weighted over mean mass, time, height, latitude and longitude. Attenuation due to



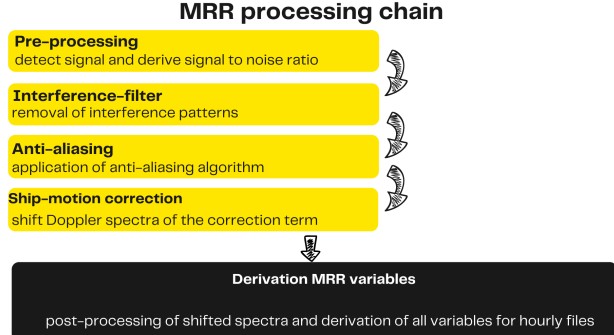

**Figure 3.** Steps of the processing applied for MRR-*PRO* data for the 1 s resolution dataset: from the manufacturer data, the pre-processing does the conversion of the Doppler velocity range and the de-aliasing, then the interference is filtered. Afterwards we apply the ship motion correction and then we derive the MRR-*PRO* variables from the shifted spectra. Finally the hourly files are merged in a daily file and the CF conventions are applied.

**Table 1.** Instruments technical specifications.

| Parameter name | MRR-*PRO* | W-band |
| --- | --- | --- |
| Operating frequency (GHz) | 24.23 | 94 |
| Operating mode | FMCW | FMCW |
| Modulation (MHz) | $0.5 - 15$ | up to 100 |
| Transmit power (W) | 0.05 | 1.5 |
| Antenna diameter (m) | 0.6 | 0.5 |
| No. of range gates | 128 | 550 |
| Range resolution (m) | 10 | 7.5, 9.2 and 34.1 |
| Resulting measuring range (m) | $0 - 1270$ | $100 - 10000$ |
| Temporal resolution | 1 s (10 s on some days) | 3 s |
| Beam width (2-way, 6 dB) | $1.5°$ | $0.48°$ |
| Nyquist velocity range (m s$^{-1}$) | $\pm 6.0$ (0 to 11.9) | 10.8, 7.3 and 5.1 |
| No. of spectral bins | 64 | 1024, 256 and 256 |
| Spectral resolution (m s$^{-1}$) | 0.1889 | 0.0415, 0.0569 and 0.0398 |
| Power (W) | 500 | 400 (Radar), 1000 W (Blower) |

precipitation has been taken into account. More details on the derivation of the MRR-*PRO* variables can be found in Garcia-Benadi et al. (2020).




**Table 2.** Chirp table definition for W-band radar.

| Attributes | Chirp Sequence (CS) | | |
|---|---|---|---|
| | CS 1 | CS 2 | CS 3 |
| Integration time (s) | 0.846 | 0.786 | 1.124 |
| Range interval (m) | 100 - 1233 | 1233 - 3000 | 3000 - 10000 |
| Range resolution (m) | 7.5 | 9.2 | 34.1 |
| Nyquist velocity ($\mathrm{ms^{-1}}$) | 10.8 | 7.3 | 5.1 |
| Doppler velocity bins | 512 | 256 | 256 |
| Doppler velocity resolution ($\mathrm{m\ s^{-1}}$) | 0.0415 | 0.0569 | 0.0398 |

## 2.3 ARM AMF-2 stabilization platform

The stabilization platform from the US-Atmospheric Radiation Measurement (ARM) program Mobile Facility 2 (AMF2) was deployed on the RV MS Merian to reduce the impact of ship motions on the Doppler zenith pointing observations (Coulter and Martin, 2016). The system, built by Sarnicola Systems, is an active stabilization system, i.e., it compensates the ship motions by adapting the position of the table surface correspondingly so that the radar stays in a zenith pointing position. It requires 120 V power and ethernet connection to a computer in a sealed container to be operational. A Hydraulic Power Unit (HPU) (the cubic metal box on the right in Figure 1 a)) must be within 600 cm of the table and weights approximately 182 kg. The HPU supplies hydraulic fluid to manipulate the length of three legs positioned below the table's surface such that the table can compensate for a large range of roll and pitch angles of the ship. More information on the stabilization platform can be obtained at https://www.arm.gov/capabilities/instruments/s-table. Ship and table motion are monitored by 2 roll/pitch sensors, one located on the ship deck and the other in the center of the table itself. A predictive computer routine uses these values to maintain the table in a geopotential level orientation at a constant height above the ships surface. Thus the table compensates for the rotational motions around the long axis of the ship (roll) and the short axis of the ship (pitch). The table did not work for approximately 35% of the time. It must be noted that the stabilization platform can compensate for the rotation of the ship but it can not compensate for the vertical movements along the vertical axis (heave, etc.) and the translations which occur because the ship rotates around its center of mass while the instruments are located elsewhere (see Section 3).

## 2.4 Motion Reference Unit (MRU) and ship reference system

When deploying a radar on a ship, vertical velocity measurements have to be corrected for ship motions, i.e. roll, pitch, yaw and heave (Figure 4). Roll and pitch variations cause the radar beam to be off-zenith and vertical range to vary with time; heave variations in time cause a vertical velocity offset and the vertical range to vary with time; the ship drift in the horizontal plane described by surge and sway may also have components in the direction of the radar beam if the radar is looking in any



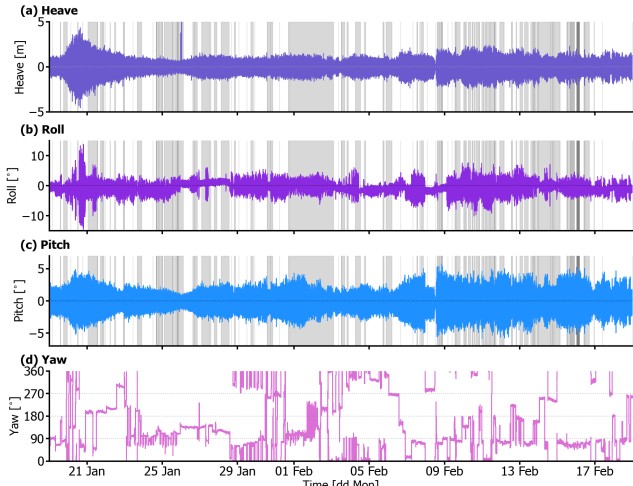

**Figure 4.** Time series of a) heave, b) roll, c) pitch and d) yaw of the RV MS Merian MRU unit during the EUREC[4]A campaign, from 19 January to 19 February 2020. Grey areas represent the periods of time in which the stabilization platform did not work.

direction tilted from the vertical relative to ocean. In this work, we will not calculate the surge and sway motions because we assume them to be negligible compared to the other terms.

All rotation angles are measured by the Motion Reference Unit (MRU) unit on the ship. The MRU is a Kongsberg 'Seapath 320' system manufactured by Kongsberg SeaTex AS. The sensor uses 2 single frequency 12 channel GPS receivers for position and heading and provides roll, pitch, yaw (heading), and heave with an accuracy of $0.03°$, $0.03°$, $0.075°$ and 0.05 m, respectively (from https://www.ldf.uni-hamburg.de/en/merian/technisches/dokumente-tech-merian/handbuch-merian-eng.pdf). The MRU is mounted at the ships center of gravity to clearly separate between translatory and rotational movements of the ship. As the radars are not at the ships center of gravity, rotational movements of the ship lead to translation of the instruments. To calculate these movements it is necessary to know the position of the instruments with respect to the MRU. They were determined as vectors in the ships coordinate system as (Figure 5 and Figure 1 c):

$$\boldsymbol{r}_{\text{W-band}} = [5.15\text{m}; 5.40\text{m}; -15.60\text{m}]$$

$$\boldsymbol{r}_{\text{MRR-PRO}} = [7.18\text{m}; 4.92\text{m}; -17.28\text{m}]$$

In the MRU sensor's conventions, roll angle is positive when port goes up, pitch angle is positive when bow goes up, and yaw angle is positive clockwise from heading angle. The heading of the ship is given as the angle clockwise from north, and refers to the x-axis of the ship system. Finally, the coordinate for heave is negative for upward directions. The angle $\eta$ is the angle between the initial position of the radar $r$ (black vector in Figure 5) and its final position $r'$ (blue vector in Figure 5) after a given motion due to the ship; $\eta$ can be decomposed in terms of roll $\theta$, pitch $\phi$ and yaw $\psi$ (Figure 5). The stabilization platform worked for 65% of the measurement time and blocked itself in a random position for the remaining 35% of the time, represented as grey areas in Figure 4. The longest interval in which the stabilization platform was not working occurred between 1 and 5 February,





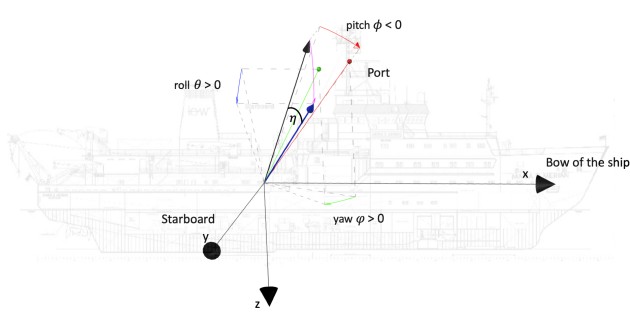

**Figure 5.** Position of the radar and its tilting due to ship motions expressed in terms of roll, pitch and yaw of the ship. The original position is represented by a black vector (arrow), and the position after the movement is given by a blue arrow. The angle representing the rotation from the initial to the final position is $\eta$ and the roll, pitch and yaw in which it can be decomposed are shown in blue, red and green, respectively. The solid red and green lines ending in filled circles of the same color represent the vector position after undergoing the rotations due first to pitch and then to yaw. The application of the rotation with respect to roll, would then bring the black array on the blue one. The sign with respect to the conventions indicated in the text is reported in the figure.

when a connection cable was badly damaged and had to be exchanged. Around 17 February, we finally fixed the stabilization platform, and in the last 2 days the stabilization platform worked continuously. Overall, we encountered the roughest sea conditions at the beginning of the campaign, and we had relatively calm sea conditions afterwards (Figure 4).

## 3   Data Processing

This section describes the corrections applied to the data to obtain the final reference dataset. Subsection (3.1) describes the drift problem in time between the radar and the ship clock that both radar datasets undergo. Synchronization of the two is thus necessary and preliminary before applying any ship motion correction. Subsection (3.2) shows how to calculate the ship motion correction term for both radar datasets and subsection (3.3) assesses the correction algorithm. Finally, subsection (3.4) shows how to filter interference for MRR-*PRO* data.

### 3.1   Tackling time drift between ship and radar time stamps

At the beginning of the campaign, we synchronized the ship and radar clock. However, the ship and radar clock cumulated a time lag $\Delta T$ that varies with time between 1 and 4 s. To calculate the time-varying $\Delta T$, we use the heave rate time series and the time series of the mean Doppler velocity averaged over the cloudy range gates $< V_d >$ of each radar profile (see section 3.3 for more details on why using the heave rate time serie). For stationary radars, i.e. radars not moving in time, the mean Doppler velocity ($V_d$) measures the mean velocity of the hydrometeors in the radar volume with respect to the radar that results as superposition of the air motion and the sedimentation speed of the drops. The average of $V_d$ over the cloud geometrical thickness $< V_d >$ fluctuates around zero in non-precipitating regions, because the sedimentation speed of cloud droplets is negligible and updrafts and downdrafts present in the cloudy column are averaged out. In precipitation regions instead (for





example in Figure 6) after 6:17:07), it becomes more and more negative, because of a larger and persistent downdraft. When the radar is moving (like on the ship), $< V_d >$ additionally tracks the radar motion (Figure 6)).

By comparing the heave rate (thin blue line) and $< V_d >$ time series (thick red line in Figure 6) we can derive the time lag $\Delta T$. Cloud droplets have a vertical speed of $w_{\mathrm{hyd}}$. The ship is moving vertically due to waves with $w^*_{\mathrm{heave}}$. The radar measures Doppler velocity $v_d$ with respect to the instrument on the ship, hence $v_d = w_{\mathrm{hyd}} + w_{\mathrm{heave}}$. Whereas $w_{\mathrm{hyd}}$ may vary with height

due to up and downdrafts in the cloud, $w_{\mathrm{heave}}$ is within one time step the same for all range gates. We average over all cloudy range gates within one time step to get $< V_d >$. By doing so we partly remove the turbulent variation of $w_{\mathrm{hyd}}$ whereas $w_{\mathrm{heave}}$ remains unaffected as it is the same for all range gates. From the ships MRU we have a time series $w^*_{\mathrm{heave}}$ , whose time stamp might be shifted against the radar time series. We calculate the variance $var(\Delta_v)$ of the difference $\Delta_v$ between $< V_d >$ and $w^*_{\mathrm{heave}}(t - Dt)$ over a time span of 20 minutes for different time shifts $Dt$. By doing so we get

$$var(\Delta_v) = var(< V_d > -w^*_{\mathrm{heave}} \cdot (Dt)) = var(< V_d >) + cov(w_{\mathrm{hyd}}, dw_{\mathrm{heave}}(Dt)) + var(dw_{\mathrm{heave}}(Dt)) \quad (1)$$

where $dw_{\mathrm{heave}}(Dt) = w_{\mathrm{heave}}(t) - w^*_{\mathrm{heave}}(t + Dt)$. Ship movement $w_{\mathrm{heave}}$ and $w_{\mathrm{hyd}}$ are not correlated, i.e. the covariance term should become zero. For an optimal timeshift $Dt$, the difference $dw_{\mathrm{heave}}$ and its variance become zero. For all other timeshifts $var(dw_{\mathrm{heave}}(Dt)))$ is positive and accordingly for the optimal timeshift $var(\Delta v_d)$ is minimal.

We then applied the resulting time lag $\Delta T$ to the ship data and interpolated this shifted series to the exact radar time,

obtaining the best correction term for each time stamp. We iterated the procedure for every radar chirp sequence since they all have different time stamps. Only after matching the time series of data from the ship and data from the radar, we could apply the ship motion correction.

### 3.2    Derivation of the ship motion's correction formula

In the following we will derive the equations to remove ship movements from the observed radar Doppler velocities with and

without a working stabilization platform. The algorithm applies to both radars. The only difference is that while for the W-band radar the correction was applied to the mean Doppler velocity, for the MRR-*PRO* the whole Doppler spectra is shifted by the correction. We will adopt bold notation for vectors i.e. $\mathbf{v} = (v_x, v_y, v_z)$ where $v_i$ are the components of the vector $\mathbf{v}$ along the various axes.

The ships coordinate system is defined by a right handed system with unit vectors $\hat{\mathbf{e}}_{\mathbf{x}}$, in direction of the bow, $\hat{\mathbf{e}}_{\mathbf{y}}$ towards

star board and $\hat{\mathbf{e}}_{\mathbf{z}}$ perpendicular to the decks downwards (Figure 5). With the ship moving in the waves this coordinate system is rotated by roll and pitch angles. This rotation is described by a rotational matrix $\mathbf{R}$ (see appendix C4). By applying the $\mathbf{R}$ on unit vectors of the ship system we get a coordinate system with $\hat{\mathbf{e}}_{\mathbf{z}}$ pointing vertically downward in the direction of earth gravitational acceleration g, and vectors $\hat{\mathbf{e}}_{\mathbf{x}}$ and $\hat{\mathbf{e}}_{\mathbf{y}}$ horizontally pointing in the direction of the ship's bow and starboard respectively. We call this system the horizontal coordinate system (Figure 5). $\hat{\mathbf{e}}_{\mathbf{z}} = [0, 0, 1]$ is the pointing direction of the $\hat{z}$

axis of the horizontal coordinate system and points downward.

The radar observes Doppler velocities relative to its own movement along its radar beam and they are positive for movements away from the radar, i.e. upward for a vertical pointing instrument. The Doppler velocity measured by the radar is

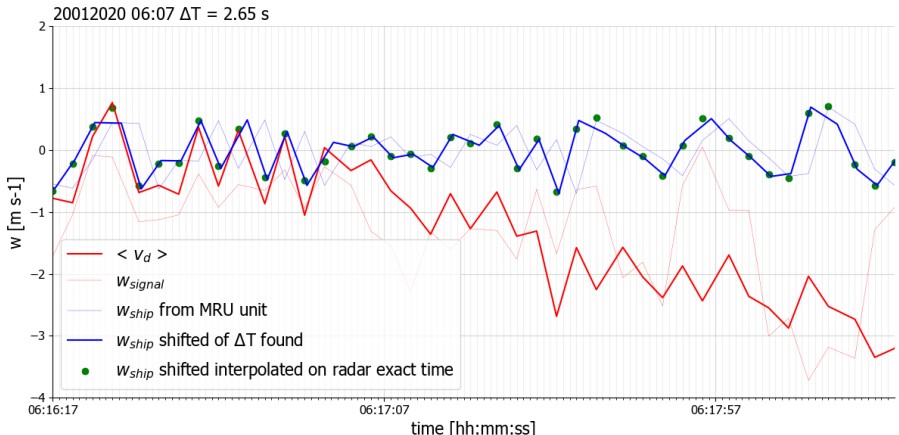

**Figure 6.** Example of time shift application calculated to obtain the best matched correction for ship motion from the 20 January 2020 over two minutes between 6:16:17 and 6:17:57 UTC. The thin blue line represents the vertical velocity measured by the ship. The thin red line is the mean Doppler velocity recorded by the radar at 1230 m, while the thick red line represents $< V_d >$, the mean Doppler velocity obtained by averaging together all cloudy pixels in each radar profile. The thick blue line represents the ship velocity after applying the time shift of $\Delta T = 2.65s$, and the green dots represents the values of $w_{ship}$ finally used for correcting for ship motion. They are obtained by cubic interpolation of the shifted ship velocity (thick blue line) on the radar time stamps. In fact, they correspond to the values in the red lines, as expected after interpolation. In the first 7 seconds of the time series, a short train of 3 waves is clearly visible that matches the radar observed mean Doppler velocity values much better after the time shift.

the projection of the particle's velocity vector on the radar line of sight. Therefore, the component of the velocity vector of the hydrometeors $w_{signal}$ measured by the radar is positive when hydrometeors move upwards. The pointing direction of the
radar in the horizontal system is denoted as $\hat{\mathbf{e}}_{\mathbf{p}}$. During times the stable table is working it is $\hat{\mathbf{e}}_{\mathbf{p}} = -\hat{\mathbf{e}}_{\mathbf{z}}$ ($\hat{\mathbf{e}}_{\mathbf{p}}$ pointing upwards, $\hat{\mathbf{e}}_{\mathbf{z}}$ pointing downwards). The velocity observed by the radar is the relative velocity between hydrometeors ($\mathbf{v}_{hydr}$) and the movement of the radar ($\mathbf{v}_{radar}$) projected onto the pointing direction of the radar ($\hat{\mathbf{e}}_{\mathbf{p}}$) that is:

$$w_{signal} = \left(\mathbf{v}_{hydr} - \mathbf{v}_{radar}\right) \cdot \hat{\mathbf{e}}_{\mathbf{p}} \tag{2}$$

where all vectors are given in the horizontal ship coordinate system and the dot represents the scalar product. The movement
of the hydrometeors can be decomposed in the horizontal system into a component along the vertical axis and one in the horizontal plane:

$$\mathbf{v}_{hydr} = v_{hydr,s}\hat{\mathbf{e}}_{\mathbf{z}} + \mathbf{v}_{wind,s} \tag{3}$$

where the term $v_{hydr,s}$ is the hydrometeor fall speed in the horizontal reference system (z component), and $\mathbf{v}_{wind,s}$ the horizontal wind vector in the horizontal reference system. Hence, we get:

$$w_{signal} = \left(v_{hydr,s}\hat{\mathbf{e}}_{\mathbf{z}} + \mathbf{v}_{wind,s} - \mathbf{v}_{radar}\right) \cdot \hat{\mathbf{e}}_{\mathbf{p}} \tag{4}$$



Now solving Eq. 4 for $v_{hydr,s}$ that is the hydrometeors fall speed in the horizontal reference system, we get:

$$v_{hydr,s} = \frac{w_{signal}}{\hat{\mathbf{e}}_{\mathbf{z}} \cdot \hat{\mathbf{e}}_{\mathbf{p}}} - \frac{(\mathbf{v}_{wind,s} - \mathbf{v}_{radar}) \cdot \hat{\mathbf{e}}_{\mathbf{p}}}{\hat{\mathbf{e}}_{\mathbf{z}} \cdot \hat{\mathbf{e}}_{\mathbf{p}}} \tag{5}$$

In the case of a working stabilization platform the radar pointing vector is exactly upwards and accordingly the scalar product $\hat{\mathbf{e}}_{\mathbf{z}} \cdot \hat{\mathbf{e}}_{\mathbf{p}}$ is equal to -1 as $\hat{\mathbf{e}}_{\mathbf{z}}$ is pointing downwards. In the limit of a non moving ship we get

$$v_{hydr,s} = -w_{signal}$$

where the opposite sign is given by the fact that the ship reference system has an opposite z direction to the one in the radar convention. Finally in the common definition with falling hydrometeors having negative velocities we get:

$$v_{hydr} = -v_{hydr,s} = -\frac{w_{signal}}{\hat{\mathbf{e}}_{\mathbf{z}} \cdot \hat{\mathbf{e}}_{\mathbf{p}}} + \frac{(\mathbf{v}_{wind,s} - \mathbf{v}_{radar}) \cdot \hat{\mathbf{e}}_{\mathbf{p}}}{\hat{\mathbf{e}}_{\mathbf{z}} \cdot \hat{\mathbf{e}}_{\mathbf{p}}} \tag{6}$$

The pointing direction of the radar $\hat{\mathbf{e}}_{\mathbf{p}}$ in Eq. 5 changes depending on whether the stabilization platform is working or not:

- if the stabilization platform is working perfectly, we assume that $\hat{\mathbf{e}}_{\mathbf{p}} = [0, 0, -1]$.

- If the stabilization platform is not working the pointing vector of the radar is moving with the ship coordinates system. Accordingly $\hat{\mathbf{e}}_{\mathbf{p0}}$ has to be rotated with the ships rotation matrix $\mathbf{R}$ in the horizontal system and we get $\hat{\mathbf{e}}_{\mathbf{p}} = \mathbf{R}^* \cdot \hat{\mathbf{e}}_{\mathbf{p0}}^{\mathbf{T}}$.
The table typically got stuck at an arbitrary position and thus the radar is pointing in an arbitrary direction. We reconstruct this direction by taking roll and pitch at time $t_0$ just before the table got stuck and assuming that the radar was pointing at this moment vertically. Orientation of the radar in the ship system is thus $\hat{\mathbf{e}}_{\mathbf{p0}} = \mathbf{R}^{-1}(t_0) * (-\hat{\mathbf{e}}_{\mathbf{z}})$, which then translates to $\hat{\mathbf{e}}_{\mathbf{p0}} = [\hat{e}_{p0x}, \hat{e}_{p0y}, \hat{e}_{p0z}] = -\mathbf{R} * \mathbf{R}^{-1}(t_0) * \hat{\mathbf{e}}_{\mathbf{z}}$ where $\mathbf{R}^{-1}$ is the inverse matrix of $\mathbf{R}$ (see Appendix C and D for more details).

The velocity vector $\mathbf{v}_{radar}$ in Eq. 2 is composed of various contributions to the motion:

$$\mathbf{v}_{radar} = \mathbf{v}_{trasl} + \mathbf{v}_{course} + \mathbf{v}_{rot} \tag{7}$$

where the velocities that add up to the radar movement are:

- **The translation velocity vector** $\mathbf{v}_{trasl}$ depends on the translation movements of the ship: heave, surge and sway (we neglect the surge and sway contribution), and it is given by

$$\mathbf{v}_{trasl} = [0, 0, w_{heave}].$$

- **The course velocity vector** $\mathbf{v}_{course}$ is due to the travel of the ship along its course, and it is given by

$$\mathbf{v}_{course} = [v_s \sin\psi, v_s \cos\psi, 0],$$

where $\psi$ is the yaw and $v_s$ is the intensity of the ship velocity vector.





– **The rotation velocity vector** $\mathbf{v}_{rot}$ describes the movement due to the rotation of the ship (roll, pitch, yaw) and the fact that the instruments are not deployed in the center of rotation but at distances $r_{\text{MRR-PRO}}$ and $r_{\text{W-band}}$ from it. Its expression is:

$$\mathbf{v}_{rot} = \frac{d\mathbf{R}}{dt} \cdot \mathbf{r}_{\text{MRR-PRO/W-band}}$$

(Appendix (C) for the derivation of the full expression).

## 3.3 Application of the correction and additional smoothing

When the table is working and the radar is pointing vertically all horizontal vector components vanish and the expression of the corrected hydrometeor velocity reduces to:

$$w = w_{signal} - \mathbf{v}_{traslz} - \mathbf{v}_{rotz} \tag{8}$$

where $v_{traslz}$ and $v_{rotz}$ are the z components of the vectors $\mathbf{v}_{trasl}$ and $\mathbf{v}_{rot}$. In this case, for calculating the velocity terms we need roll, pitch, heave rate. All these data are provided by the ship navigation system. Angles roll and pitch are necessary because the rotation of the ship moves the radar vertically. Course (heading and speed) are not necessary as the they are horizontal components not seen by the vertical looking radar.

When the table is not working, the pointing vector of the radar is most of the time not vertical and may have a horizontal component (scalar products of $\hat{\mathbf{e}}_{\mathbf{p}}$ with horizontal vector components do not vanish). Accordingly course of the ship and horizontal wind may contribute to the signal. We therefore need additional parameters heading and speed of the ship and the horizontal wind above. The first two are provided by the ships navigation system. For the wind we used the output of dedicated ICON simulations run (Klocke, personal communication) over the EUREC[4]A domain to extract the horizontal wind profiles at the closest time and place of the ship, corresponding to the time when the table was not working. This type of correction affected 35% of the total measurement time. The low time resolution of the model output compared to the observations made this correction less accurate than the one applied for the 65% of the data in which the stabilization platform worked correctly.

An example of the application of the correction for ship motions when the stabilization platform is working is visible in Figure 7 a)-d), for the case of the 20 February 2020 between 2:22 and 2:27 Local Time (LT), with especially strong waves (compare with Figure 4). When comparing the original (Figure 7 a)) to the corrected mean Doppler velocity field (Figure 7 b)), one can quickly notice that many of the intense and frequent vertical bars disappear, providing a more homogeneous and continuous field. However, the correction cannot entirely remove the disturbances, as shown in Figure 7 b) by some visible vertical bars remaining despite the correction. Some possible reasons for the mismatch observed are the distance between the MRU sensor and the radar equipment, especially considering that we measured by hand the radar equipment's position. We are also assuming that the stabilization platform keeps the radar perfectly in zenit, but it is hard to quantify the accuracy of such hypothesis a little error in the zenith alignment can produce disturbances. Moreover, the time lag quantification $\Delta T$ can be unprecise due to the reinitialization of the chirp generator of the W-band radar. Such time is random and adds an unknown uncertainty to the time stamps assigned to the measurements. Finally, the coarse temporal resolution of the MRU data makes an interpolation to the radar data necessary. Fig 6 nicely shows the rapidly changing $w_{ship}$ which misses due to its coarse

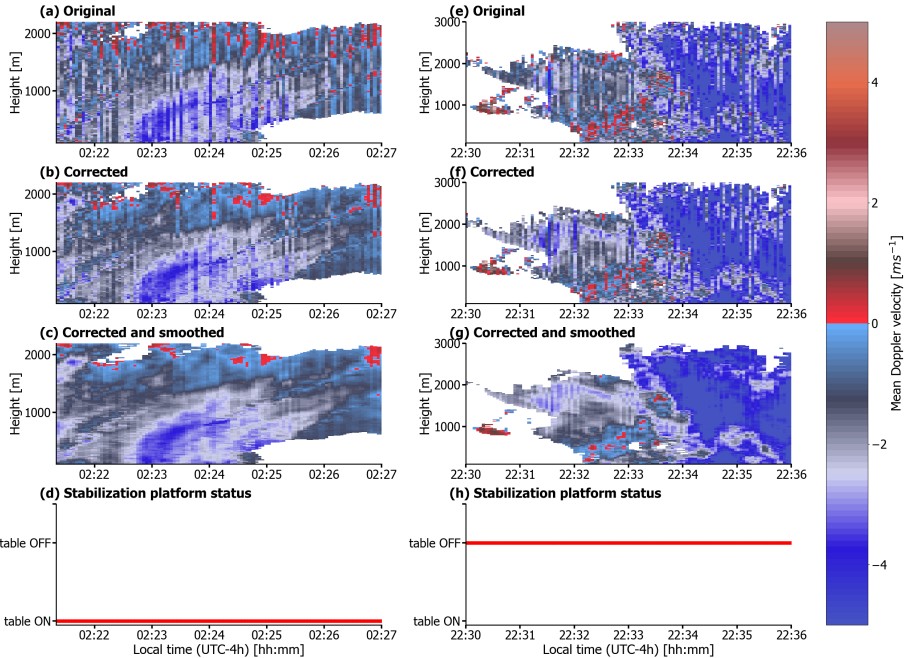

**Figure 7.** On the left, example of ship motion correction algorithm applied when the stabilization platform was working, on the 20 January 2020 from 2:21 to 2:27 LT (UTC-4h) for the W-band radar data with de-aliasing applied: a) Original mean Doppler velocity field without any correction algorithm applied, b) Mean Doppler velocity after application of the correction from ship motions, c) Mean Doppler velocity after application of correction from ship motions and smoothing (running mean over 9 s), and d) status of the stabilization platform. On the right, another example when the stabilization platform did not work, on the 12 February 2020 between 22:30 and 22:36 LT: e) Original mean Doppler velocity field without any correction algorithm applied, f) Mean Doppler velocity after application of the correction from ship motions, g) Mean Doppler velocity after application of correction from ship motions and smoothing (running mean over 9 s) and h) status of the stabilization platform.

temporal resolution the real minima and maxima making an interpolation challenging. We applied a running mean over three time stamps (i.e., over a 9 s time interval) to account for these limitations (Figure 7 c)). The final signal obtained shows an almost continuous field in mean Doppler velocity.

Figure 7 d)-h) show the correction performance when the table did not work, on the 12 February between 22:30 and 22:36 LT. Even if the final smoothing (Figure 7 g)) improves the $v_d$ pattern compared to the field obtained when applying the correction algorithm only (Figure 7 f)), overall the performance is worse than in Figure 7 a)-c) and vertical stripes are markedly visible in all fields. We applied the time shift and the ship motion correction also to the MRR-*PRO* data obtaining similar results as will be shown in the section 3.4.

Figure 8 for one hour shows the impact of the time matching and of the correction applied to the signal in the frequency space. The heave rate is the main contributor to the vertical motion of the radar (Figure 8 a)), as described by looking at a cloudy range gate. The rotational components are approximately one order of magnitude smaller because the instruments are

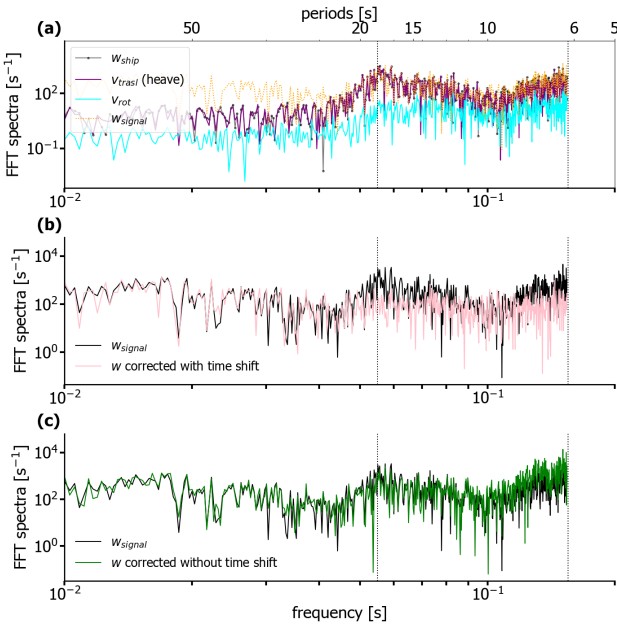

**Figure 8.** a) Fast-Fourier Transform (FFT) of the ship vertical velocity and of its translational (purple) and rotational (cyan) components for one hour of data collected at a cloudy range gate located at 1605 m from the radar. The ship velocity highlights two main wave periods around 6 and 17 seconds, indicated by dashed vertical lines. In the total ship velocity (black dotted line), the contribution of the rotational component is minor compared to the contribution of the translational component (heave). b) Comparison of the FFT of the radar uncorrected mean Doppler velocity (black) and of the FFTs of the corrected mean Doppler velocity with time shift applied. The two main wave peaks disappeared in the corrected signal (pink line). c) Same as middle panel, but with the corrected mean Doppler velocity calculated without applying the time shift. In this case, the wave frequencies are not removed from the corrected signal (green line).

not too far from the center of mass of the ship, and the rotation of the ship moves the instrument not much along the vertical.
For this reason, it represents the vertical velocity of the ship due to the waves, as previously stated. The frequencies of the waves at approximately 6 and 17 s (highlighted by the vertical dashed bars in Figure 8) are visibly removed in the FFT spectra of the corrected mean Doppler velocity only if the time shift is applied (compare Figure 8 b) and c)). Finally, the increase of the spectra towards the Nyquist frequency, between 0.1 and 0.5 Hz, indicates that there are higher frequencies above 0.5 Hz, folded back into this interval. Such frequencies do play a role that is not resolved by MRU nor by the radar itself. The final
smoothing over the 9 s time window removes the high frequency components and it is thus crucial to obtain a better signal to noise. However, the 9 s smoothing degrades the average horizontal resolution: given $V_{ship, mean}$ the mean ship speed, the degradation would change from $V_{ship, mean} * 1$ s to $V_{ship, mean} * 9$ s. For an average ship speed of 3 ms$^{-1}$, the resolution would change from 3 m to 27 m, resulting in a slightly higher resolution than the vertical 30 m one. However, daily maximum speeds for the ship can reach also 9 ms$^{-1}$, producing thus a coarser resolution.





### 3.4 Removal of Interference patterns and correction for ship motions for MRR-*PRO* dataset

The MRR-*PRO* electronics interfered with the ship instrumentation and with the stabilization platform electronics during the whole campaign. To be able to use the data collected, we removed the interference patterns using a noise removal mask. The interference draws periodical disturbances with peak intensity decreasing with height. Since the interference peaks are larger than the mean noise level calculated using the Hildebrand-Sekkon method by the manufacturer's processing (Hildebrand and Sekhon, 1974), multiple small peaks appear in the MRR-*PRO* spectra. The mean Doppler velocity and the spectral width of such noise spectra are random, depending on which noise peak is the highest (Figure A1).

The MRR-*PRO* dataset produced by the software of the manufacturer is initially processed with the MRR-*PRO* post-processing tool developed by Albert Garcia-Benadi (publication in preparation) (Pre-processing and anti-aliasing steps in Figure 9). The algorithm allows to obtain de-aliased Doppler spectra over a physically realistic Doppler velocity range. For data with 5 or 10 s integration time, this tool is sufficient to remove the interference pattern. No ship motion correction can be applied to those data because their integration time is larger or similar to the typical wave period (see Figure 8 a)) and Doppler variations due to heave motions are smoothed out. We resampled the data collected from the 19 to the 25 January 2020 with 5 s to a 10 s integration time, to reduce the impact of ship motions completely.

The post-processing of the data collected with 1 s integration time, i.e. from the 25 January to the 19 February 2020, is more complex (see Figure 3). For the 1 s resolution data, the tool from Garcia-Benadi cannot remove the interference patterns as it did for the 10 s integration time dataset. Hence, to obtain Doppler spectra without interference we applied a noise removal mask (Interference filter in Figure 9) based on specific conditions:

1. We calculated for each spectrum the prominence of all its spectrum peaks, i.e. each peak's ability to stand out from the surrounding baseline of the signal. Then, we derived the difference between the maximum and minimum prominence and calculated their difference ($\Delta P$). The difference is tiny for spectra containing only interference patterns and no signal from hydrometeors, while it is significantly larger for a Doppler spectrum detecting hydrometeor backscattering (see for reference on Figure A1). Spectra affected by interference patterns were removed by selecting spectra with $\Delta P > 1$ $mm^6 mm^{-3}$, where the threshold value of one was determined empirically;

2. In addition, we posed a condition on the spatial continuity of mean Doppler velocity (mdv) in the lowest 600 m. The mdv obtained from spectra affected by interference shows very large random absolute values. Doppler spectra detecting hydrometeors produce continuous mdv field in space. We discarded all profiles where the difference of consecutive mdv values along the profile shows more than 8 abrupt peaks (threshold decided empirically).

3. We apply a spatial filtering to remove spurious noisy pixels: the filters excludes all pixels where $\Delta P > 1$ that have less than 3 adiacent neighbours fullfilling the same condition.

It is almost impossible to distinguish the signal due to hydrometeors from the one due to interference in the attenuated reflectivity field $Z_{ea}$ of the original dataset. However, after applying the noise removal mask the hydrometeor signal becomes clearly visible (Figure 9 a) and b)). The time correction (see Section 3.1) and post-processing tool from Garcia-Benadi were then ap-

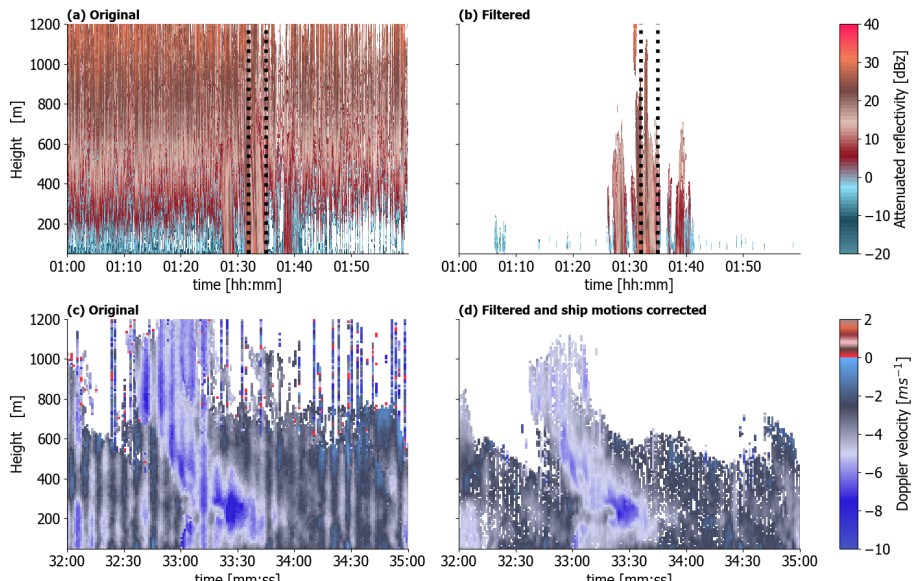

**Figure 9.** a) MRR-*PRO* attenuated reflectivity on the 13 February 2020 at 1:00 UTC after manufacturer processing, without any additional interference filtering or ship motion correction. b) Same as a), but with the noise removal mask applied. c) MRR-*PRO* mean Doppler velocity without any correction over a 2 min time interval selected from the time interval shown in a) between 1:32 and 1:35, and b) and highlighted with dashed black lines. Vertical stripes structures are visible due to ship motions. d) Same as c) but with interference removed and time shift and ship motion corrections applied to the data. The striped structure present in c) almost entirely disappeared and made a hook rain structure visible, possibly caused by downdraft wind mixing.

plied to remove the time lag $\Delta T$, de-alias and obtain a physically realistic Doppler velocity range for all Doppler spectra above the noise level (Anti-aliasing in Figure 9). Then, ship motion correction derived with the calculations presented in section 3.2
is applied and all the main MRR-*PRO* variables of interest are derived from the corrected Doppler spectra. Figure 9 c) and d) show a dynamic vortex structure not visible in the original data that emerged from the noise after applying the correction on the mean Doppler velocity field.

## 4 Characteristics of trade wind cumulus clouds and precipitation

To give an overview of the meteorological conditions encountered on each of the 32 days of campaign, Table 3 lists the daily
mean atmospheric temperature (T2m), rain rate (RR), liquid water path (LWP), relative humidity (RH) and pressure (P) for each day of the campaign. They are collected at the radar base, which is approximately 20 m above sea level.

Figure 10 shows that the vast majority of the encountered liquid clouds have a LWP smaller than $100\ \mathrm{gm}^{-2}$, with a median value of $11\ \mathrm{gm}^{-2}$ in agreement with Schnitt et al. (2017), who sampled the region between 10 and 20 ° N and -40 and -60 ° W in December 2013. Noise and gain drifts in the passive channel of the radar lead to positive LWP retrieved values even

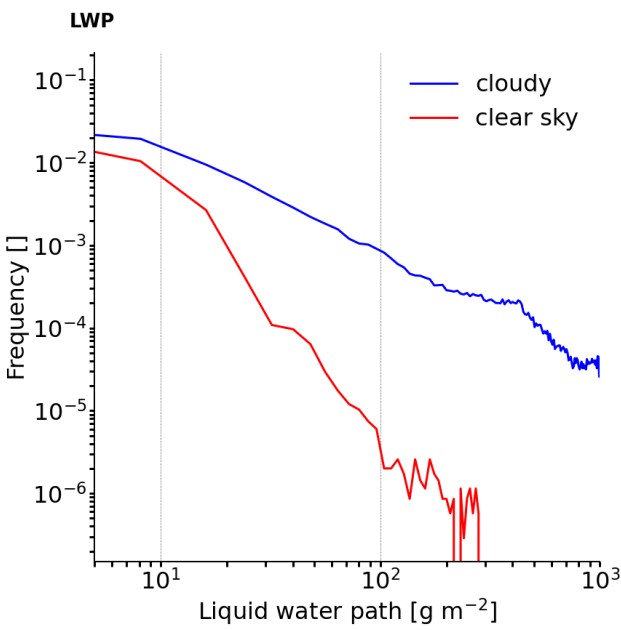

**Figure 10.** LWP distribution for cloudy (blue) and clear-sky (red) conditions, for the whole campaign.

in clear sky conditions. These data have median and standard deviation of 1.2 and 5.4 $\text{gm}^{-2}$, respectively, which are within
the retrieval uncertainty (about 30 $\text{gm}^{-2}$). The standard deviation of the clear sky distribution can be considered as a sort of
uncertainty of the LWP values retrieved with the neural network algorithm, and can be used to correct the LWP values, as done
in Jacob et al. (2019).

We compared the obtained IWV values with the IWV retrieved from GNSS by Bosser et al. (2021). The mean of the IWV
retrieved from W-band radar single-channel retrieval is 31.7 $\text{kgm}^{-2}$, the median is 32.3 $\text{kgm}^{-2}$ and the standard deviation of
the distribution is 5.15 $\text{kgm}^{-2}$. The bias between the mean value of the IWV distribution from W-band and the IWV distribution
from GNSS is 3.4 $\text{kgm}^{-2}$, which is consistent with the bias estimated with ground-based GNSS stations reported in Figure 9
of Bosser et al. (2021). The spread between the GNSS and the radar derived values of IWV can be due to the strongly varying
bias component that affects the GNSS IWV estimations from MS Merian (Bosser et al., 2021), as well as to limitations in the
IWV single channel retrieval.

To show the full potential of the collected radar dataset, we display one case study of an extended precipitating cloud field
occurring on 12 February 2020 from 15:30 to 17:00 UTC in the trade wind alley at about °13.5 N and °57 W. On that date,
the ship encountered a cloud system identifiable as a flower type using the corrected reflectances from Moderate Resolution
Imaging Spectroradiometer (MODIS) TERRA (Platnick et al., 2003), with a diameter between 200 and 250 km that generated
precipitation during the afternoon (Figure 11 c)). When comparing the signals observed by the W-band radar (Figure 11 a)) and
the MRR-*PRO* (Figure 11 b)), the different sensitivities of the two instruments become evident; while the W-band is capable of
detecting cloud and precipitating hydrometeors, the MRR-*PRO* is sensitive to larger raindrops only. The interference patterns


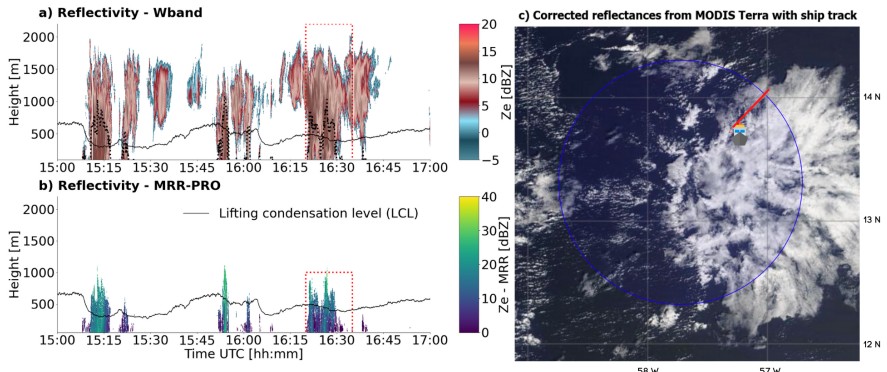

**Figure 11.** Overview of the selected case study of the 12 February 2020 between 15:00 and 17:00 UTC: a) Radar reflectivity from W-band radar. The black solid line represents the lifting condensation level calculated using the surface variables, while the black dotted line shows the highest detected signal from the MRR-*PRO*. The red dashed box represents the area shown in Figure 12. b) Attenuated reflectivity from MRR-*PRO*. c) Ship track on top of the corrected reflectance from MODIS Terra, displaying the flower cloud occurring in the area. The corrected reflectance from MODIS is a product that uses the bands 3 (479 nm, red), 6 (1652 nm, green), and 7 (2155 nm, blue) produced in near real-time (NRT) to provide natural looking images. The blue circle represents the orbit of the HALO aircraft. The image is taken at 16:03 UTC. Source:https://observations.ipsl.fr/aeris/eurec4a/Leaflet/index.html

reduced the ability of the MRR-*PRO* to detect precipitation in a way that it is difficult to quantify. The W-band radar system collected echos in the first 2200 m showing the complex internal structure of the clouds. The cloud base detected from the W-band radar ranges between 750 and 1250 m, and does not exactly correspond with the LCL values (black solid line in Figure 11 a), b)) obtained for this case, while in non-precipitating conditions LCL is higher. Cloud top ranged between 1700 and 2100 m.

The high vertical resolution mode for the W-band radar (7 m up to 1230 and 9 m from 1230 m to 3000 m) detected distinctive features in the radar moments (Figure 12 a)-d)). Slanted filaments of higher reflectivity between 16:25 and 16:30 at around 1000 m (Figure 12 a)) suggest a correlation of the size of the drops with air motions; Figure 12 b) displays clear areas in the cloud where larger mean Doppler velocities are associated with heavy rain. The spectral width field (Figure 12 c)) also benefits from the high temporal and spatial resolution. It shows patterns that suggest a correlation between large reflectivities and mean Doppler velocity values to large spectral width values. Finally, the skewness field shows patches of positive and negative skewness emerging from the noise. Further analysis of the Doppler spectra in precipitation is necessary to interpret these patches and exploit the skewness signatures (Acquistapace et al., 2019).

Also, for the MRR-*PRO*, the high vertical and time resolution allowed to reveal relevant structures in the lower precipitation field. Despite the small gaps caused by the filtering of the interference, the reflectivity (Figure 12 e)) shows a general decrease of the Ze values with decreasing altitude possibly due to evaporation and/or shear, except for two moments, around 16:22 and 16:26 UTC, when the high observed Ze values correspond to high fall speeds of -4 ms$^{-1}$ close to the surface (Figure 12 f)). The case study highlights a large variability of fall speeds in the lowest 300 m, possibly connected with sub-cloud layer dynamics.

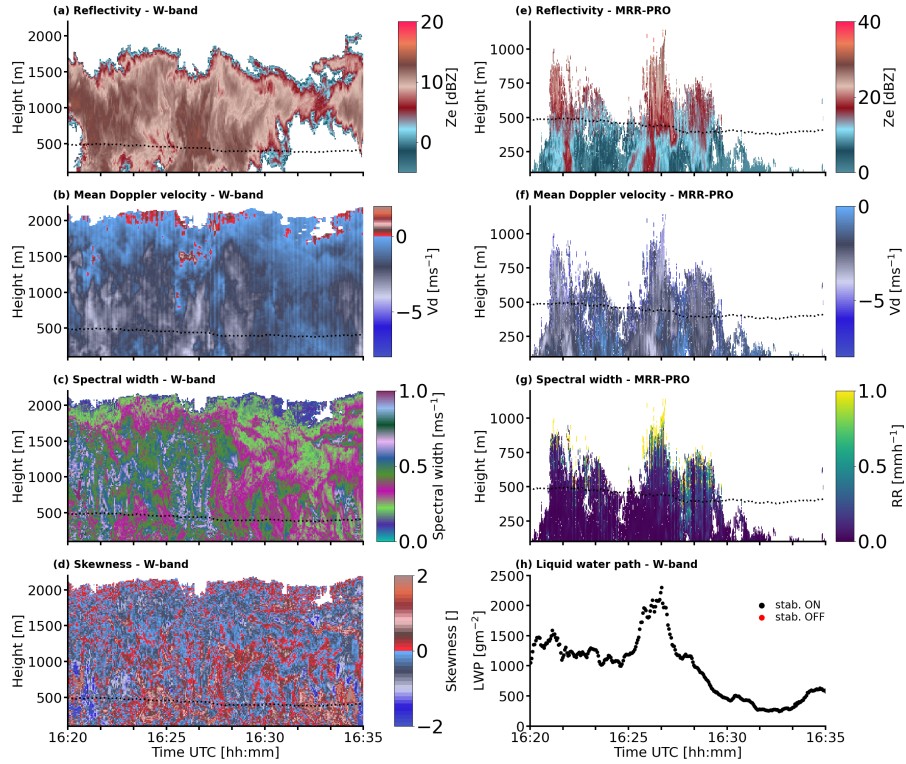

**Figure 12.** Detail of the case study shown in Figure 11 for W-band variables a) Reflectivity, b) Mean Doppler velocity, c) Spectral width, d) Skewness and for MRR-*PRO* variables e) Attenuated reflectivity, f) mean Doppler velocity, g) Spectral width. LWP and stabilization platform status are displayed in h).

The fall speed field can also trace such dynamics, as the vortex structure in Figure 9 d). In contrast with the reflectivity and fall speed fields, the rainfall rate does not show a substantial variation during the selected case study (Figure 12 g)). During the case study the stabilization platform worked continuously and the LWP registered very high values under rainy conditions. Note that values above $1000 \text{ gm}^{-2}$ should not be considered as reliable because of contamination due to rain.

## 5 Data availability

The data presented in this paper can be accessed at AERIS repository and on the ARM database in NetCDF format, under (https://doi.org/10.25326/235) (Acquistapace et al., 2021c). This DOI was assigned to the new version of the dataset, produced after fixing a bug in the standard post-processing script and correcting the LWP neural network dataset. In the dataset:

– technical radar variables were removed and stored in hourly technical files that can be accessed upon email request to the manuscript's corresponding author.

– A Doppler velocity variable has been added to facilitate the usage of the Doppler spectrum variable.



   – A compact data version suitable (including only radar moments) for the EUREC[4]A intake catalog has been produced following CF conventions. In the EUREC[4]A book now available at https://howto.eurec4a.eu, some example codes on how to read the data and plot basic quantities are available for users. More support will be added here in the future.

Auxiliary dataset used for correcting for ship motions can be found at https://doi.org/10.25326/156 (Acquistapace et al., 2021b). together with the outdated version of the radar data.

   The MRR-*PRO* data can be accessed at AERIS repository and on ARM databases (https://doi.org/10.25326/233) (Acquistapace et al., 2021a). Data are organized in daily NetCDF files. Also this dataset is following CF conventions and is included in the EUREC[4]A intake catalog with example codes for basic operations with the data.

Finally, the code used for post-processing the W-band radar data is published on github (https://github.com/ClauClouds/w-radar). The post-processing software for MRR-*PRO* data will be published in a devoted publication (Garcia-Benadi, 2021, in preparation). The code for ship motion correction and interference filtering and for deriving the plots of the paper can be accessed at zenodo https://doi.org/10.5281/zenodo.5014089. All the data are visualized in hourly and daily plots on the quicklook browser https://bit.ly/3xLkb9b. For improving the data visualization (Zeller and Rogers, 2020), we created color 420   palettes using the Colorgorical tool (Gramazio et al., 2017), and we used them for all plots of the quicklook browser as well as for many of the graphics of this publication.

## 6   Lessons learned

As underlined in the introduction, we experienced various challenges deploying active remote sensing instruments on the MS Merian research vessel. For encouraging and facilitating future deployments on ships, we collected some issues we encountered 425   that future technological developments could solve.

   The ship motion correction algorithm described here has also been tested on the radar data collected on the Meteor research vessel, where the ship navigation system data used 0.1 s (10 Hz) time resolution. We noticed that increasing the time resolution of the ship position data from 1 Hz to 10 Hz is beneficial for the ship motion correction. Spectral analysis of the data from MS Merian indicates that there are components of the ship movement at frequencies above 0.5 Hz which had to be filtered out by 430   a simple gliding average smoothing operator. We therefore recommend using 10 Hz for future campaigns.

   A significant limitation to the exactness of the correction came from the need to synchronize the radar clocks with the GPS time from the ship. This is necessary to assign the right correction to be used for the measurements. The synchronization problem is a well-known issue for aircraft measurements, and more research is needed to tackle this point. At least for ship purposes, a possible solution could come from including a high-resolution sensor in the radar that can tell the radar inclination 435   and heave for each radar partial chirp sequence time stamp with high precision. Currently W-band radar position data (inclination, and elevation) are provided with the time resolution of the total sampling time, which is approximately 3 s and is too poor for an effective correction of ship motions.

   Finally, we strongly recommend a preliminary test phase to the campaign, as we did in this work. The time spent before the campaign in testing the instruments allowed us to take care and solve small details that could have strongly affected



the measurement quality, like the vibration of the pole or the best setup for the computer connections onboard the RV. We experienced interference problems on the ship. Interference is always hard to detect and to solve, but we recommend making some test measurements with all instruments and checking the raw data obtained. In our case, the MRR-*PRO* interference was not visible on the quicklooks of the control system, but it significantly impacted the observations. In a test phase, interference could be tackled and possibly solved.

**7   Conclusions**

This paper presents the W-band and MRR-*PRO* dataset collected on the MS Merian research vessel during the EUREC[4]A campaign between 19 January and 19 February 2020. We installed and operated two radars on the stabilization platform deployed on the ship with the collaboration of the ARM AMF2. The suite of instruments constituted an advanced setup for studying the precipitation life cycle in the tropical region and the first deployment of Doppler instrumentation on the RV MS

Merian. The ship sampled a broad oceanic region between 6° N and 13.8° N and 60° W to 51° W. The data collected provide a precious characterization of the trade region and the transition from the trades to the intertropical convergence zone. The ship sampled some mesoscale oceanic eddies, and the observations will provide vital information to understand the impact of sea surface heterogeneities on the marine boundary layer.

We developed an algorithm to correct the Doppler observations from ship motions and successfully applied it to the W-band

dataset. The algorithm initially calculated the time shift between the radar time stamps and the ship navigation system time to identify the radar position with respect to the motion reference unit as best as possible. It then applies the correction term to the mean Doppler velocity. For the MRR-*PRO* data, in addition to the ship motion correction algorithm, we also developed advanced post-processing techniques to filter out interference problems between the MRR-*PRO* and the stabilization platform. We first removed the interference pattern, and we then applied the correction directly to the Doppler spectra. Then, we used

the standard post-processing to derive the moments and the other rain-related variables from the corrected Doppler spectra.

The corrected fields remove most of the typical striped pattern due to heave motion in the mean Doppler velocity (W-band) and fall speed (MRR-*PRO*) that ship motions cause to Doppler measurements. The correction for ship motion was applied to the entire dataset. However, for 35% of the data, the stabilization platform did not work. We corrected this data subset using the horizontal wind profile extracted from NWP ICON-LEM model runs and horizontal ship velocity.

A unique feature of the dataset is the high temporal and vertical resolution; the time resolution is 3 s (W-band) and 1 s (MRR-*PRO*). Below 3000 m, i.e., where most of the cumulus liquid clouds develop, the range resolution of the W-band is 9 m or 7 m, while the one of the MRR-*PRO* is 10 m. The profiles of the W-band radar moments detected with unprecedented detail showed characteristics patterns that will be explored in future works, especially for what concerns the spectral width and skewness. MRR-*PRO* variables like the fall speed contain important detailed information on the dynamical evolution of the

rain in the sub-cloud layer and its interaction with the dynamics. We exploited the passive 89 GHz channel available on the W-band radar to retrieve LWP in cloudy conditions and IWV in clear-sky situations. The LWP retrieval is a neural network retrieval provided by the radar manufacturer, while the IWV is derived from a single-channel quadratic regression between





the 89 GHz brightness temperatures obtained in clear-sky and the IWV measured by the radiosoundings launched at the exact times. We assessed the IWV retrieval by comparing it to the IWV estimations obtained by GNSS (Bosser et al., 2021). We

found a bias of 3.4 kgm$^{-2}$, in agreement with what reported in Bosser et al. (2021).

The high resolution of the collected datasets and the possibility of synergies with the other instrumentation on board, i.e., Raman lidar, wind lidar, cloud kite, make the described observations a benchmark dataset for future analysis as model studies and evaluations, comparing satellite retrievals and process studies. We made the data public and accessible on the AERIS and the ARM database platforms to achieve these purposes. Moreover, we also made the data accessible online via the EUREC[4]A

intake catalog, and hourly and daily quicklooks are available online for browsing into the data.

Future work will focus on improving the quality of the correction: when the wind lidar corrected dataset from MS Merian will be published, it will provide horizontal wind profiles for the entire campaign and thus allow to obtain a better correction for the 35% of the dataset collected when the stabilization platform did not work.

*Video supplement.* The corresponding author realized a short video from the campaign that was approved by the ship board and is available

online at the following link: https://www.youtube.com/watch?v=EdWNS77qMNA

## Appendix A: LWP retrieval using neural network

This appendix describes the neural network retrieval developed to retrieve LWP from the single passive channel at 89 GHz of the W-band radar, exploiting radiosoundings launched in the region of the campaign. The dataset consisted of the 3 radiosonde stations collocated in Grantley Adams International Airport (Barbados), International airport of "Le raizet" (Guadaloupe) and

Piarco international Airport (Trinidad) and the one location from the ERA-Interim reanalysis (see Table A1). All available profiles from Jan 1994 to Dec 2016 were used. In total, there were 41588 profiles. We used 29111 (70%) randomly chosen profiles for the ANN training and 10% of the dataset for the validation. We used the remaining 20% for the retrieval evaluation (test dataset). For each profile, we calculated the LWP following Löhnert and Crewell (2003). A radiative transfer model was used to simulate TB values at 89 GHz. The absorption of oxygen and water vapor was calculated according to Rosenkranz

(Rosenkranz, 1998, 1999). The absorption by liquid water was calculated using the Rayleigh scattering approximation and the model from Liebe et al. (1991) and Liebe et al. (1993).

We used as input variables for the ANN training the simulated brightness temperatures (TB), day of the year, near surface temperature, relative humidity, and pressure. The values of temperature, relative humidity, and pressure closest to the surface were taken from profiles. The calculated LWP was used as the target variable. The input and the target variables were normal-

ized using the min-max function. The ANN consists of two layers: a hidden layer with 5 neurons and an output layer with one neuron. The hyperbolic tangent is used as an activation function for all neurons. The standard error backpropagation algorithm was used for the training. After the training, we evaluated the retrieval using the test dataset. The retrieval root mean square



error (RMSE) is 33 $\text{gm}^{-2}$. During the radar operation, the ANN uses TBs measured by the passive channel and measurements of the surface temperature, relative humidity, and pressure from the weather station.

**Appendix B: Calculation of the wind speed in the ship reference system $\mathbf{v}_{wind,s}$**

In the Earth reference system, the horizontal wind vector in absolute coordinates is given with the zonal component towards East, and the meridional component towards North. If it has a speed $v_{wind,E}$ and a direction indicated by $\alpha$ with respect to the North, it can be written in Cartesian coordinates as:

$$\mathbf{v}_{wind_E} = [-v_{wind,E}\sin\alpha, -v_{wind,E}\cos\alpha, 0] \tag{B1}$$

After applying the rotational matrix of ship motions, the ship coordinate system (see Fig. 5) has:

- x axis to the heading of ship, horizontal, perpendicular to the gravity acceleration g,

- y axis to the starboard of ship, right side, horizontal, perpendicular to g,

- z axis downward in direction of g

If $\psi$ is indeed given relative to heading, the equations describing the wind in the ship reference system are:

$$
\begin{aligned}
\mathbf{v}_{wind,s} &= [u_{ship}, v_{ship}, 0] \\
&= v_{wind,E}[-\sin(\alpha-\psi+90), \cos(\alpha-\psi+90), 0] \\
&= v_{wind,E}[-\cos(\alpha-\psi), \sin(\alpha-\psi), 0]
\end{aligned}
\tag{B2}
$$

because the ship coordinate system is rotated clockwise by $\psi-90$, and the y-axis of the ship has opposite direction with respect to the Earth reference system.

We can hence re-write Equation 6 as:

$$\mathbf{v}_{hydr} = [0, 0, w] + v_{wind,E}[-\cos(\alpha-\psi), \sin(\alpha-\psi), 0] \tag{B3}$$

**Appendix C: Calculation of the rotation vector $\mathbf{v}_{rot}$**

Let's define $\eta$ the rotation angle resulting from ship motions. The rotation matrix $\mathbf{R}^*$ associated with a generic rotation $\eta$, is the product of the rotation matrices associated with the roll, pitch and yaw, in the way the angles provided by the MRU sensor on the ship are defined. The prescribed order for the MS Merian is roll ($\theta$), pitch ($\phi$), heading (yaw) ($\psi$), heave. The general expression for the rotation matrix is hence given by:

$$\mathbf{R}^* = \mathbf{CBA}$$


where $\mathbf{A}$ is the rotation matrix for the roll, $\mathbf{B}$ is the rotation matrix for the pitch, and $\mathbf{C}$ is the rotation matrix for the yaw. The expressions for $\mathbf{A}$, $\mathbf{B}$, $\mathbf{C}$ are:

$$\mathbf{A} = \begin{pmatrix} 1 & 0 & 0 \\ 0 & \cos\theta & -\sin\theta \\ 0 & \sin\theta & \cos\theta \end{pmatrix} \tag{C1}$$

$$\mathbf{B} = \begin{pmatrix} \cos\phi & 0 & \sin\phi \\ 0 & 1 & 0 \\ -\sin\phi & 0 & \cos\phi \end{pmatrix} \tag{C2}$$

$$\mathbf{C} = \begin{pmatrix} \cos\psi & -\sin\psi & 0 \\ \sin\psi & \cos\psi & 0 \\ 0 & 0 & 1 \end{pmatrix} \tag{C3}$$

The expression for $\mathbf{R}^*$ is:

$$\mathbf{R}^* = \begin{pmatrix} \cos\psi\cos\phi & \sin\phi\sin\theta\cos\psi - \sin\psi\cos\theta & \sin\phi\cos\theta\cos\psi + \sin\psi\sin\theta \\ \sin\psi\cos\phi & \cos\psi\cos\theta + \sin\psi\sin\phi\sin\theta & -\sin\theta\cos\psi + \sin\psi\sin\phi\cos\theta \\ -\sin\theta & \cos\phi\sin\theta & \cos\phi\cos\theta \end{pmatrix} \tag{C4}$$

The heading term $\psi$ is necessary only when the stabilization platform gets stuck and we ignore it when stabilization platform works. We call $\mathbf{R}$ the rotational matrix obtained when neglecting $\psi$, that applies for 65% of the data. Rotational movement
of the ship leads to translational movement of the instrument because it is not located in the center of mass of the ship. The location of the radar with respect to the center of mass at any moment of the rotation is $\mathbf{r}_{rot} = \mathbf{R} * \mathbf{r}_{radar}$. Its velocity is the derivative with respect to time : $\mathbf{v}_{rot} = d/dt(\mathbf{R}*\mathbf{r}_{radar}) = d\mathbf{R}/dt*\mathbf{r}_{radar}$, with x, y and z the coordinates of the radar location vector on the ship.

The velocity variations of the stabilizing system with respect to the MRU are described by the derivative with respect to time
of the vector $\mathbf{r}_{rot} = \mathbf{R} \cdot \mathbf{r}_{radar}$:

$$\mathbf{r}_{rot} = \begin{pmatrix} x\cos\phi + y\sin\phi\sin\theta + z\sin\phi\cos\theta \\ y\cos\theta - z\sin\theta \\ -x\sin\theta + y\cos\phi\sin\theta + z\cos\phi\cos\theta \end{pmatrix} \tag{C5}$$

The vector $\mathbf{r}_{rot}$ does not contain $\psi$ because we ignore heading when table is working. Adopting the point as a symbol for the temporal derivative, the rotational velocity results in:

$$\mathbf{v}_{rot} = \begin{pmatrix} -x\dot\phi\sin\phi + y(\dot\phi\cos\phi\sin\theta + \dot\theta\sin\phi\cos\theta) + z(\dot\phi\cos\phi\cos\theta - \dot\theta\sin\phi\sin\theta) \\ -y\dot\theta\sin\theta - z\dot\theta\cos\theta \\ -x\dot\theta\cos\theta + y(\dot\theta\cos\phi\cos\theta - \dot\phi\sin\phi\sin\theta) - z(\dot\theta\sin\theta\cos\phi + \dot\phi\sin\phi\cos\theta) \end{pmatrix} \tag{C6}$$





## Appendix D: Calculation of $\hat{\mathbf{e}}_{\mathbf{p0}}$ and $\hat{\mathbf{e}}_{\mathbf{p}}$

When the stable table stops working it leaves the table and thus the instrument in an arbitrary orientation denoted by a fix vector $e_{p0}$ in the (rolling and pitching) ship system. This vector can be transformed to the horizontal system by multiplication with rotation matrix $\mathbf{R}^*$ (see Appendix C) as $\hat{e}_{p(t)} = \mathbf{R}^* * e_{p0}$:

$$\hat{\mathbf{e}}_{\mathbf{p}} = \mathbf{R}^* * [\hat{e}_{p0x}, \hat{e}_{p0y}, \hat{e}_{p0z}] \tag{D1}$$

$\mathbf{e}_{\mathbf{p0}}$ can be calculated from the position in which the table was when it got stuck. The stabilization platform angles at the time $t_0$ when the table got stuck can be obtained as follows. For the roll

$$\theta_{tbl,S}(t_0) = \theta_{ship}(t_{final}) - \theta_{tbl,S}(t_{fin}), \tag{D2}$$

for the pitch

$$\phi_{tbl,S}(t_0) = \phi_{ship}(t_{final}) - \phi_{tbl,S}(t_{final}) \tag{D3}$$

and for the yaw:

$$\psi_{tbl,S}(t_0) = \psi_{ship,}(t_{final}) \tag{D4}$$

where $\theta_{tbl,S}(t_{fin})$, $\phi_{tbl,S}(t_{final})$ and $\phi_{tbl,S}(t_{final})$ are the last recorded positions of the stable table relative to the ships deck as recorded in the raw data files of the stabilization platform. $t_{final}$ is the closest time of the ship time serie to the time $t_0$ when the table got stuck, in which there was a record of ship data. The point vector $\hat{\mathbf{e}}_{\mathbf{p0}}$ can then be obtained as:

$$\mathbf{e}_{\mathbf{p0}} = \mathbf{R}^{*-1}(t_0) * [0, 0, -1] \tag{D5}$$

where the inverse rotational matrix $\mathbf{R}^{*-1}$ is calculated as:

$$\mathbf{R}^{*-1} = \mathbf{A}^{-1} \cdot \mathbf{B}^{-1} \cdot \mathbf{C}^{-1},$$

where $\mathbf{A}^{-1}$, $\mathbf{B}^{-1}$ and $\mathbf{C}^{-1}$ are the rotational matrices associated to the roll, pitch and yaw angles of the table at the time $t_0$: $\theta_{tables|_{t_0}}$, $\phi_{tables|_{t_0}}$ and $\psi_{tables|_{t_0}}$. The expressions of the matrices $\mathbf{A}^{-1}$, $\mathbf{B}^{-1}$ and $\mathbf{C}^{-1}$ can be obtained from the expressions of $\mathbf{A}$, $\mathbf{B}$, $\mathbf{C}$, by using negative angles.

The final expression for the pointing direction is:

$$\hat{\mathbf{e}}_{\mathbf{p}} = \mathbf{R}^* \cdot \mathbf{e}_{\mathbf{p0}} = \mathbf{R}^* \cdot \mathbf{R}^{*-1}(t_0) * [0, 0, -1] \tag{D6}$$

where $\mathbf{R}^{*-1}$ is provided by the definitions above.



**Appendix E: Calculation of $\mathbf{v}_{course}$ and $\mathbf{v}_{trasl}$**

The course vector $\mathbf{v}_{course}$ is determined by the ship velocity $\mathbf{v}_s$ and its heading $\psi$. We decided to calculate the ship velocity by deriving the UTM coordinates given by the MRU-GPS system of the ship with respect to time. We hence get:

$$\mathbf{v}_{course} = [v_s \sin\psi, v_s \cos\psi, 0] \tag{E1}$$

The translation vector $\mathbf{v}_{trasl}$ that the ship undergoes has three components:

  – heave: it is the variation of the z position due to the waves and it is provided by the MRU system. Its projection along the radial beam might be of the order of the hydrometeor fall speed.

  – surge and sway are the short term variations of the position in the x/y direction compared to the ship velocity slowly
varying term. They are not provided by the MRU system but can be derived from the ship velocity data by applying a short time averaging. We will neglect their contribution, since it should be small as long as the point vector $\hat{\mathbf{e}}_{p0}$ does not deviate more than $10°$ from the vertical direction.

We can then write:

$$\mathbf{v}_{trasl} = w_{heave} \cdot \hat{\mathbf{e}}_{\mathbf{z}} = [0, 0, w_{heave}] \tag{E2}$$

where $\hat{\mathbf{e}}_{\mathbf{z}}$ is the unit vector $\hat{\mathbf{e}}_{\mathbf{z}} = [0, 0, 1]$ and the heave velocity results in being positive downwards.

*Author contributions.* Claudia Acquistapace took care of the data curation, the funding acquisition and the project administration for the deployment of the instruments on the ship. She also developed the software used for the post-processing and prepared the manuscript original draft. Jan H. Schween was involved for methodology and conceptualization of the post-processing algorithms; Nils Risse and
Giacomo Labbri were involved in the data visualization. Alexander Myagkov helped to specify observational settings for the W-band radar, applied and described the LWP retrieval developed by RPG, and assisted in checking W-band radar data. Albert Garcia Benadi was involved in the programming and the execution of the MRR-*PRO* post-processing while Rosa Gierens took care of the programming and the execution of the standard W-band radar data processing. Susanne Crewell initiated the deployment of the instrumentation and provided constructive feedbacks on the manuscript structure and organization; Richard Coulter provided supporting algorithms to operate the stabilization platform
and covered a supervision role in the execution of the mentioned codes. All co-authors reviewed and edited the manuscript to finalize it for publication.

*Competing interests.* No competing interests are present.



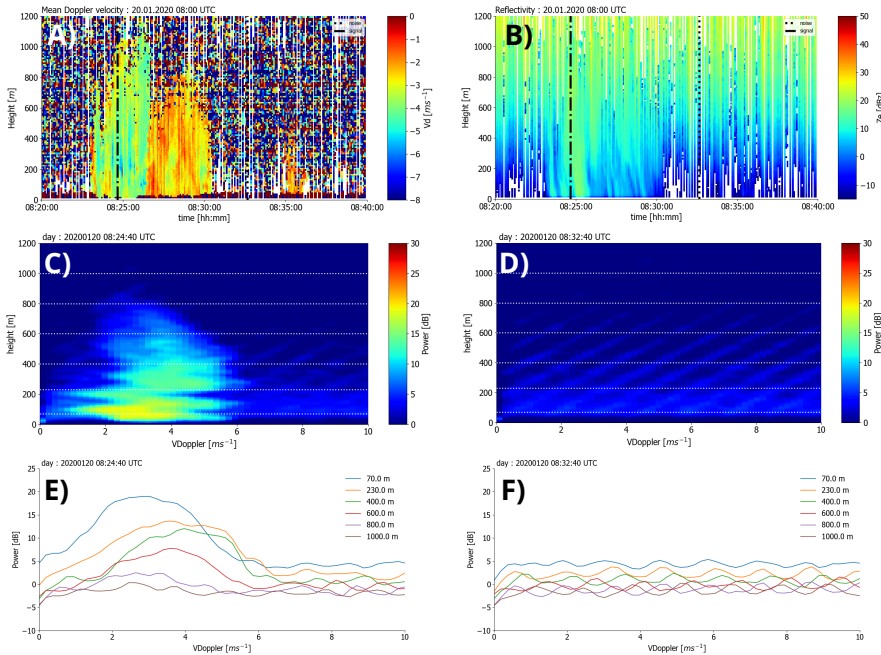

**Figure A1.** Interference in the MRR-*PRO* data: a) mean Doppler velocity for one selected hour. The black vertical lines correspond to the selected times for plotting the spectra shown in panels C to F. b) Same, for reflectivity. c) Height spectrogram of MRR-*PRO* Doppler spectra collected at 8:25:40 UTC during rain, where the horizontal white dashed lines indicate the heights selected for plotting the spectra shown in e) and f). d) same for Doppler spectra collected at 8:32:40, in the noise. E) Doppler spectra collected along the vertical selected profile for rain. F) Doppler spectra collected along the vertical selected profile for noise.

*Acknowledgements.* This work was funded by the Deutsche Forschungsgemeinschaft (DFG – German Research Foundation) under the Research Grants Programme - Individual Proposal with title "Precipitation life cycle in trade wind cumuli", project number 437320342, https://gepris.dfg.de/gepris/projekt/437320342.


We gratefully acknowledge the support by the SFB/TR 172 "ArctiC Amplification: Climate Relevant Atmospheric and SurfaCe Processes, and Feedback Mechanisms (AC)$^3$" funded by the DFG (Deutsche Forschungsgemeinschaft).

This article is partially based upon work from COST Action PROBE, supported by COST (European Cooperation in Science and Technology), www.cost.eu.


We want to thank Rainer Haseneder-Lind, the ARM team, and Steven M. Bormet, for their fantastic support and collaboration in the testing phase of the installment of the stabilization platform on the ship in Emden (DE), for the help and the patience provided after the campaign, for the correction of the data and the delays in the upload on the ARM database. We also thank Jun. Prof. Heike Kalesse-Los and Johannes Rötthenbacher for deploying the MRR-*PRO* on the MS Merian and for the the fruitful discussion on in the development phase of the correction algorithm. We want to thank Annika Daehne for the administrative support she provided across the different phases of the campaign.


We also would like to acknowledge the ship crew for the brilliant support offered in the installation of the equipment onboard MS Merian





and for facing all the technical issues encountered during the campaign. We thank Daniel Klocke, for running ICON simulations that were used for correcting the radar data from ship motions. Finally, we thank Markus Ritschel for the fruitful discussions onboard MS Merian on how to implement the correction for ship motions, and the scientific crew onboard MS Merian for the collaborations developed onboard, with a special thanks to Prof. Eberhard Bodenschatz for finally fixing the stabilization platform. We thank Juan Antonio Bravo Aranda and

Lukas Pfitzenmaier for the work done for developing the post-processing radar Matlab software tool that was used in this work.



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



**Table 3.** Daily mean values of the main surface variables observed on the MS Merian during the EUREC$^4$A campaign: T2m is the air temperature 2m above the radar base, which is approximately 20 m above sea level, RR is the rain rate. The liquid water path (LWP) is derived from the collocated 89 GHz channel microwave radiometer and RH and P are the relative humidity and air pressure from a weather station positioned next to the radar equipment.

| DAY | T2m [$°C$] | RR [mmh$^{-1}$] | LWP [gm$^{-2}$] | RH [%] | P [hPa] |
|---|---|---|---|---|---|
| 19.01.2020 | 26.35 | 0.0 | 1. | 63.6 | 1013.9 |
| 20.01.2020 | 25.95 | 0.57 | 30. | 72.4 | 1013.3 |
| 21.01.2020 | 26.85 | 1.0 | 71. | 67.2 | 1011.7 |
| 22.01.2020 | 27.25 | 1.42 | 0. | 63.2 | 1010.3 |
| 23.01.2020 | 26.85 | 0.99 | 12. | 69.2 | 1009.7 |
| 24.01.2020 | 26.15 | 0.57 | 318. | 76.2 | 1010.4 |
| 25.01.2020 | 26.85 | 0.67 | 13. | 67.5 | 1011.9 |
| 26.01.2020 | 26.65 | 0.0 | 23. | 67.4 | 1012.2 |
| 27.01.2020 | 26.95 | 1.37 | 391. | 75.2 | 1012.0 |
| 28.01.2020 | 27.25 | 0.0 | 50. | 74.9 | 1010.8 |
| 29.01.2020 | 27.15 | 0.32 | 26. | 72.4 | 1010.9 |
| 30.01.2020 | 27.55 | 0.0 | 20. | 71.5 | 1011.7 |
| 31.01.2020 | 27.35 | 0.0 | 8. | 70.0 | 1012.7 |
| 01.02.2020 | 27.45 | 0.31 | 13. | 64.8 | 1013.0 |
| 02.02.2020 | 27.45 | 0.49 | 6. | 62.3 | 1012.0 |
| 03.02.2020 | 27.05 | 0.0 | 3. | 68.2 | 1013.3 |
| 04.02.2020 | 27.25 | 0.0 | 8. | 69.2 | 1013.1 |
| 05.02.2020 | 27.05 | 0.45 | 10. | 68.3 | 1014.1 |
| 06.02.2020 | 27.15 | 0.0 | 11. | 65.7 | 1013.9 |
| 07.02.2020 | 26.75 | 1.77 | 31. | 63.7 | 1013.9 |
| 08.02.2020 | 26.55 | 7.43 | 35. | 65.8 | 1013.6 |
| 09.02.2020 | 26.85 | 1.38 | 4. | 66.3 | 1015.3 |
| 10.02.2020 | 26.65 | 0.80 | 106. | 67.3 | 1015.1 |
| 11.02.2020 | 26.55 | 0.30 | 54. | 68.7 | 1014.6 |
| 12.02.2020 | 26.35 | 0.43 | 85. | 70.3 | 1014.1 |
| 13.02.2020 | 26.55 | 0.70 | 51. | 68.5 | 1012.7 |
| 14.02.2020 | 27.35 | 3.24 | 53. | 67.3 | 1012.9 |
| 15.02.2020 | 26.85 | 0.89 | 22. | 68.0 | 1012.1 |
| 16.02.2020 | 26.75 | 1.06 | 47. | 70.2 | 1011.3 |
| 17.02.2020 | 27.05 | 0.33 | 15. | 68.6 | 1011.9 |
| 18.02.2020 | 26.75 | 4.22 | 312. | 71.4 | 1013.2 |
| 19.02.2020 | 26.05 | 0.62 | 319. | 74.1 | 1013.1 |





**Table A1.** Information about the radiosonde stations (data taken from https://ruc.noaa.gov/raobs/intl/) and the ERA-Interim data point used for the retrieval of LWP using neural networks.

| Data type | Station name | Station number | Station lat/lon |
|---|---|---|---|
| radiosonde | BB-GRANTLEY-ADAMS-INTL | 00078954 | 13.040 / -59.290 (z = 56.0 m) |
| radiosonde | MF-LE-RAIZET-GUADELOUP | 00078897 | 16.160 / -61.310 (z = 8.0 m) |
| radiosonde | TD-PIARCO-INTL-AIRPORT | 00078970 | 10.370 / -61.210 (z = 15.0 m) |
| ERA-Interim | BB-ERA-Barbados | 30401103 | 12.750 / -59.250 (z = 12.5 m) |