# Peer review of "EUREC4A's Maria S. Merian ship-based cloud and micro rain radar observations of clouds and precipitation."

_Earth System Science Data, 2021_

## Author Comment (AC1)

REPLY TO RC1
MANUSCRIPT ESSD-2021-265

The manuscript describes the set-up, attitude control, and exemplary measurements of two vertical pointing radars onboard RV MS Merian during Eurec4a. Main focus of this data paper is the compensation of the ship motion in situations where the active stabilization platform was properly working and different treatment of data in situations where the platform got stuck in arbitrary orientations.

The manuscript is consistent and well written and certainly deserves publication as data paper. There are only minor modifications required. I like the lessons learned section, I hope that this will be considered in future campaigns.

We thank the reviewer for the attentive and constructive review of the publication.

Some general comments:

Obviously, the authors did develop the methods behind the ship motion correction by themselves, without borrowing from the airborne radar community (e.g. Bange, J. et al., 2013: Measurement of aircraft state and thermodynamic and yynamic variable, in: Airborne Measurements for Environmental Research: Methods and Instruments, edited by: Wendisch, M. and Brenguier, J-L., https://doi.org/10.1002/9783527653218.ch2). The methods are the same and there is an agreement between the two worlds (air – sea).

Based on the reviewer's suggestions, we expanded the literature review on the topic, exploring the Bange et al 2013 text suggested ( https://onlinelibrary.wiley.com/doi/10.1002/9783527653218.ch2 ) which is also part of the book "Airborne Measurements for Environmental Research: Methods and Instruments" by M. Wendish and J-L. Brenguier ( https://onlinelibrary.wiley.com/doi/book/10.1002/9783527653218) with particular interest in the chapter 9 on LIDAR and RADAR observations. However, we did not find specific mentions on corrections for mispointing on a moving platform. Therefore, we decided to add a general sentence as follows: "Similar methods have been derived for airplane based measurements with Doppler measurements see e.g. Bange et al, 2013".

For the W-band radar you do not mention anything about attenuation (gaseous, liquid) which certainly has to be considered, you call it just reflectivity (factor). Whereas for the MRR you talk about attenuated equivalent reflectivity (factor). This should be consistent. Otherwise one might assume that data from W-band radar are corrected for gaseous and liquid attenuation as well as for Mie effects. Reflectivity factor z implies Rayleigh approximation. However, since both systems are mm-wave systems you better write about effective reflectivity factor. This implies that Mie scattering effects have to be considered in the interpretation of reflectivity factor.

Thank you for this comment. The reflectivity provided by the postprocessing of the Wband radar( 94 GHz, so λ is 3.19 mm)  data is the equivalent reflectivity factor, i.e. the reflectivity calculated from the measured returned power assuming that the target is composed of liquid water droplets whose diameter is less then one tenth of the radar wavelength (droplets are treated as Rayleigh scatterers). When drops are larger than 3 mm, which is often the case in the data we collected, this approximation is not true and the equivalent reflectivity factor differs from the reflectivity factor. On top of that, while the RPG manual In (https://www.radiometer-physics.de/downloadftp/pub/PDF/Cloud%20Radar/RPG-FMCW-Instrument_Manual.pdf ) details the gas loss correction, liquid attenuation affects the estimation of the equivalent reflectivity factor provided. All Wband radars as the one deployed on the ship (from RPG) provide in their data the equivalent reflectivity factor without corrections for liquid attenuation and Mie scattering because always Rayleigh approach is applied and we published the data following this line of conduct. Subsequent scientific publication planned on precipitation will focus on this aspect and provide the correction for Mie and liquid absorption for Wband radar data and compare them properly ( on same resolution in time and space) with MRR data.

For the MRR data, the postprocessing algorithm calculates the equivalent radar reflectivity taking into account the attenuation due to liquid water contained in raindrops as well as the equivalent reflectivity non attenuated. For the MRR case, it is more difficult to get out of the Rayleigh approximation for the data collected, due to the different wavelength, but by definition, the equivalent reflectivity factor is assuming Rayleigh approach.

The plots show the equivalent reflectivity without any correction applied, so they are coherent with respect to the displayed quantity. We clarified better the differences between the datasets regarding the Rayleigh hypothesis.

I think the labels/numbers in the figures are too small, but I leave this to the technical editor.

We also wait for the technical editor's opinion on this. Thank you for noticing this aspect.

Reply to minor comments:
Line 14: I think DOI's have to be given in the abstract, but this is a task for the technical editor
We also do not know what is the best practice here. We wait for technical editor comments.

Line 22: What does OA stand for?
OA stands for ocean-atmosphere and is an acronym that represents the ocean-atmosphere research component that was part of the EUREC4A campaign (https://eurec4a.eu/overview/eurec4a-oa/). We included it because the RV Maria S. Merian was deeply involved in the operations regarding the investigation of sea-air interactions and the mesoscale eddies impact on the boundary layer. We included in the text a description of the OA and the link.

Line 69: Section instead of Session
Thank you for the comment, corrected.

Line 78: can you give here some numbers about the temporal drift

We added a estimation of the drift time. However, the drift is variable in time. We modified the sentence as follows: " Despite this effort, the time stamp synchronization suffered from a drift of the clocks with respect to the Global Positioning System (GPS) time of the ships inertial system variable between 1 and 4 s that we had to consider in the correction of the data for ship motions. "

Line 127: Why longitude and latitude with the same temporal resolution as the radar data are not copied to the radar dataset?
The latitude and longitude data come from the ship because they haven't been recorded by the radar. The lat and lon data have been resampled on the radar time resolution in the daily file version which is published online. Daily files are supposed to be used from the largest amount of users because they are ncdf files in a standardized format, easy to read and contain all the meteorological variables of interest.
Lat and lon are not included in the hourly radar files because such files contain only the radar-specific information, on top of the radar variables, and are of interest for specific radar applications only.

Lines 170 - 175: can you also give the temporal resolution of the MRU?
We added that the temporal resolution of the MRU is 1 s, thank you for the comment. The sentence reads now as follows: "All rotation angles are measured by the Motion Reference Unit (MRU) unit on the ship with a time resolution of 1 s."

Lines 179 - 184: this should go to section 2.3
Thank you for the comment: the sentence was somehow doubled, so we added some parts in section 2.3 and we left here only the sentence related to the figure 4. Please check the resulting text in the diff_versions.pdf file attached.

Figure 5 is hardly readable, the sketch of MS Merian is 2D, whereas the vectors are 3D; this figure should be improved considerable
We thank the reviewer for these comments. The readability has been improved by enlarging the fonts. We modified the image by splitting it in two parts. On one side one image of the RV, on the other we added the 3D reference system.

Lines 206 + 207: "whereas w_heave … gates" is repeated from above
Thank you for the comment. We stated it again because we wanted to clarify why by averaging over height the w_heave does not change. However, we removed the second sentence since your suggestion indicates that it is redundant.

Lines 224 - 230: confusing, ex,ey,ez is first ship relative and later horizon relative, maybe you could use different notations for the two reference systems
We thank the reviewer for the comment. Since the ship reference system and the horizontal reference system differ only because of the z direction, we preferred to keep the same notation, and just specify the direction. We went through the text and found an inaccuracy in the reference to the reference system. We corrected that and we hope that now the text can be less confusing.

Line 226: appendix C or equation C4

Corrected, thank you!

Please find here the new version of the plot:

This image has been substituted to the old Figure 6

Horizontal reference system here. Ep is expressed with respect to the horizontal coordinate system

We changed the naming

Unfortunately there is no way to our knowledge to measure bending and twisting of the ship body itself. Also, We assumed that such deformations could be neglected because we did not experience rough sea conditions.

The figure was modified, thanks for the comment. We also extended the color bar range, to avoid the saturation for high Vdoppler values (see comment below)

corrected thank you

In this plot, we chose to represent a precipitating shallow cumulus cloud not extending above the inversion and a deeper cumulus with top above 3000 m, to represent the variability in precipitation that we observed during the campaign. The high fall velocities observed are generated by the deeper cloud core, and the unrealistic effect mostly come from the fact that we used the same color bar for Doppler velocities even if the observed velocities span different ranges. A modified version of the plot has been prepared to avoid the saturation in the vd more negative values.

Line 311: signal to noise "ratio"
corrected

Line 313: Table 2 gives 7.5 to 34 m resolution. For the horizontal resolution also beamwidth has to be considered
Values presented in the table are provided by the manufacturer and refer to range vertical resolution.
We included in the table that is vertical resolution.

Line 321: numbering Fig. A1 is confusing. The figure does not belong to Appendix A
Thanks for the comment. However, the numbering of the images is ruled by the copernicus latex template and is assigned automatically, to my knowledge. The ordering of the images is independent from the ordering of the appendixes. We hope that the technical editor can help in this respect.

Figure 9 caption: interpretation "and made a hook rain structure … wind mixing" should not go to caption
thank you for the comment, we moved the sentence in the main text and riformulated the sentence at line 350 as follows: "Figure 9 d) shows a hook rain structure visible, possibly caused by downdraft wind mixing. The vortex structure was not visible in the original data (Figure 9 c)) and emerged from the noise after applying the correction on the mean Doppler velocity field. "

Figure 12: I think, images from both radars should have the identical height axis range and identical color bar range and color map. This makes comparisons between W- and K-band much easier, even though it is not the objective of the paper.
We thank the reviewer for this comment. The goal of this paper is not to compare between the W and K band radars, but instead to display the measurements collected and show their potential. The high resolution adopted with the Wband radar can be incredibly beneficial for process studies and model evaluations, therefore we opted for showing it visibly. More detailed studies on precipitation that are planned by the main author will exploit the diverse information on precipitation coming from the usage of the W and K band frequencies. There the approach suggested by the reviewer will be exploited for the future analysis.

Line 399: It might be worth to discuss shortly the observed differences between both radars and how the complement each another. Different attenuation due to different wavelengths, different sensitivity, …

In line with the previous reply, we think that an extensive discussion on this point would perfectly fit in a scientific publication focusing on precipitation more than in a data paper whose goal is to present a dataset, that can be used in various different ways, not necessarily exploiting the multifrequency approach. We actually will include such discussion in the planned publication on precipitation.

Line 434: just a comment: some airborne systems (unfortunately not HALO/HAMP) have an IRS/IMU as close as possible to the radar antenna
interesting thank you :)

Line 507: … by alpha clockwise from North
corrected thank you

Line 507: for non-meteorologists you could add "(the direction where the wind is coming from)" this makes it easier to understand the minus signs in Eq. B1
The sentence has been modified as follows: " In the Earth reference system, the horizontal wind vector in absolute coordinates is given with the zonal component towards East, and the meridional component towards North and it represents the direction where the wind is coming from."

Line 514: If "yaw" psi is indeed …
added, thanks

Line 523: heave is not discussed here
We removed the information on heave collected as last, because as suggested, is not relevant here.

Line 531 till end of page: check order of description of r_rot and v_rot, looks like repeating of definitions
We thank the reviewer for this comment, we rearranged the paragraph in a more linear way, we hope. Thank you for noticing.

Line 544, Eq. D1: should be e_p0 = … (?)
we added the definition of E_po

Line 547, Eq. D2: why not t_final for theta_tbl,S
Thanks for the comment. The distinction between t_final and to, the time when the table got stuck, follows from the fact that the time to is measured by the stable table, and not by the ship time. The t_final time stamp is the closest time to at which the ship sensor collected observations of roll, pitch, yaw. There might be a small difference between these two times because they are measured by two different sensors. This is why we maintain the separation.

---

## Author Comment (AC3)

REPLY TO RC2
MANUSCRIPT ESSD-2021-265

Acquistapace et al. describe in this paper the deployment, operation and data processing of two radar systems (W-band cloud radar and micro rain radar) on the research vessel Maria S. Merian during the EUREC4A campaign in early 2020. They describe the setup of the radars on a stabilization platform and the correction of measured Doppler velocities, both for times when the stabilization platform was operational and when it was stuck in an arbitrary position.
The manuscript is well written and clearly structured. The data set is easily accessible.
I have a some comments and would recommend publication after addressing those.

We thank you the reviewer for the nice description of the paper.

General comments regarding the data:

It would be nice, if the data for ship motion was published along with the latest version of the radar data. Now, users have to access different data sets. And this also adds the danger of possibly using outdated radar data that are published along with the ship motion data (https://doi.org/10.25326/156) and could confuse some users.
Is there a way to access ship location information to use together with the hourly data? What would users need to do if they want to know where the Maria S. Merian was located for a specific hourly data file?

The data for ship motion that have been included in the old dataset version contain the terms that we calculated for deriving the correction to the mean Doppler velocity, namely for each radar bin, the Ep vector, the time delay, the rotational velocity, the translational velocity, the course velocity and the wind velocity in the ship reference system. We also stored the correction term obtained from those and the denominator in the correction formula (see Formula 5). We did not include them in the latest version of the dataset because such data are not supposed to be used, and they could have created more confusion. They do not include the ship position, which is published on the AERIS data portal and publicly available at 1s resolution at this link https://observations.ipsl.fr/aeris/eurec4a-data/SHIPS/RV-MARIASMERIAN/dship/msm_089_1.tsg

To facilitate the data usage, we included the lat/lon information on the position of the ship in the daily dataset files (https://observations.ipsl.fr/aeris/eurec4a-data/SHIPS/RV-MARIASMERIAN/wband_radar/final_dataset/daily_intake/), where the lat/lon have been interpolated on the radar time resolution (3 s). The same was done also for the MRR data (https://observations.ipsl.fr/aeris/eurec4a-data/SHIPS/RV-MARIASMERIAN/mrr-pro/), but in this case the data were interpolated on 1s time resolution.

The dataset was created with the following concept: daily files (for both MRR and Wband) are the easiest to open, visualize and contain all the radar variables and the ship position that are of immediate interest for users. They are the main way to access the data and pick the case studies of interest. Once identified the exact hour of interest, radar interested users can open the specific hourly file if they are interested in deep advanced radar variables. In fact, all radar moments are stored in the daily file, and the hourly radar files contain in

addition only radar specific variables (full Doppler spectrum, sensitivity limit, mean noise level). For this reason, they are only for specific radar use. We hope to have clarified better the data structure.

Specific comments regarding the manuscript:
Figure 3: it is a bit unclear to me, what this figure should convey. The processing steps that are shown in the figure are not easily understandable by just looking at it. A lot of necessary information is given in the figure caption. One could think about adding these additional information from the caption also into the figure itself. The order in which the steps are mentioned in the caption does not match the order in the figure. I would suggest to either expand this figure a bit so that the processing steps are understandable by only looking at the figure or to remove the figure and expand the respective text section to thoroughly explain the processing steps. In the current state, the processing is a bit hard to follow between the figure, the caption and the text.
We thank the reviewer for the provided suggestion, which we took since we totally agree with the expressed opinion. We removed the figure and we expanded the text to describe the MRR data processing chain.

Line 157: "computer in a sealed container", what is the meaning behind this? What does the computer do? Is the information necessary? Also, does it have any impact that thecontainer is sealed?
The computer is controlling the stable table behavior. It is positioned in a sealed contained because, since it is standing outside, right under the stable table, it might be damaged by the atmospheric conditions. The information is necessary for future deployments, to properly install the equipment. The container does not impact the functioning of the computer.

Section 2.3: What are the limits of the stabilization platform compensation? I.e. maximum roll and pitch angles that can be compensated? Was this relevant during this campaign?

The report on the stabilization platform provided by Coulter et al, 2016 (https://doi.org/10.2172/1253916) describes the behavior of the table. In particular, the dimensions of this table allow for ±30° of pitch and ±25° roll, but as other tables, such extremes cannot be reached simultaneously (see Figure 2 in the report). The report also describes how the table responds to different sea conditions. Figure 4 shows that the observed conditions during our campaign were in the range of the conditions sampled in the testing phase, documented in the report. Therefore, we are confident that the table could work in a manageable regime of sea motions during EUREC4A.

Figure 5: this figure is hard to read in the current state. The ship's drawing in the background distracts from the coordinate axes plotted on top and is not necessary to understand the different rotation angles when the figure is first mentioned. I understand, that this figure is referenced later as well as an example on how the coordinate system is defined with respect to the ship. Could this figure be split into two figures for the different purposes?
We agree with the comment of the reviewer and we modified the figure

Line 193: "... time lag DeltaT that varies with time between 1 and 4 s" is the variation

systematic (e.g. linear increase over the measurement duration)? Or are these random variations?
The variations we observed are quite random.

Line 214 and following: was the time correction done for each time step individually or was this applied to longer intervals?
The time correction was done for each time step individually. In particular, for each chirp time step. Within one time stamp of the radar, that takes 3 s, there are intermediate chirp times. The correction was applied to each chirp time stamp.

Line 373: "... cloud system identifiable as a flower type ..." It might be helpful for the reader to add a reference describing the different cloud organization names here.
Added, thanks for the comment.

Line 513: what is psi? If this is the yaw angle and the difference between psi and heading can be something other than zero, I don't understand why heading and yaw are used interchangeably in other parts of the manuscript (e.g. l167 (p9, second paragraph, third line), l523).
Psi is the yaw, and the difference between psi and heading is none. We have been erroneously using two terms because people use both. We will stick to yaw and remove the heading in the whole manuscript. Thanks for the comment.

Line 547, line 552: what is t_fin? Should it be t_final?
Yes, thank you, it is a typo and we corrected.

Figure A1: The vertical lines in A) and B) are only barely discernible or not at all. Maybe using another colormap (something like the ones used for the other figures in the manuscript) would help? The authors switch between lower case and upper case letters for the different figure parts between the figure itself and the figure caption and even within the figure caption.
Thank you for the comment, we updated the figure based on your suggestions.

Technical corrections
Line 8: hydrometeors -> hydrometeor
corrected
Line 33: time -> temporal
corrected
Line 35: setup -> setups
corrected
Line 127: add lat, lon as coordinates to the text
corrected
Line 188: I don't understand the use of the word "preliminary" here.
Removed, thanks for noticing

Line 475: "in agreement with what reported in" -> "in agreement with what was reported in"
corrected
Table 3: unit [deg C] not italic
corrected

---

## Author Comment (AC4)

REPLY TO RC3
MANUSCRIPT ESSD-2021-265

The manuscript introduces the radar data sets collected during the EUREC4A campaign between trade wind region and tropical convergence zone on the research ship Maria S. Merian. The data set is unique and of high interest for the community. Necessary post-processing and some derived products published alongside the manuscript are presented. This manuscript deserves publication after correction of quite a number of small weaknesses and inaccuracies in the presentation.
Questions to the editorial office rather than the authors arise from the use of webpage referencing and a reference to an unpublished manuscript.

We thank the reviewer for recognizing the importance of the dataset collected and we try of best to answer the comments and solve the inaccuracies and weaknesses found in the text.

General points:

The introduction and beginning of the section 2 left me confused about what to expect in this paper. Please be specific about what you will provide in this data set as early as possible. In the manuscript it only becomes clearer step by step. IWV is first mentioned in the beginning of section 2. LWP somewhat later, before IWV is detailed again. I would suggest to mention all these in the introduction and add a product table of all data set, their sampling rate, their expected accuracy, etc.

We included the derivation of the IWV estimations in the introduction. The modification can be found at line 31 in the new version of the manuscript: "The 89 GHz passive channel available in the W-band radar system allowed to characterize the columnar amounts of liquid water and integrated water vapor was retrieved only in clear sky conditions by means of a linear regression with co-located radiosoundings.". We refer to all the data products in section 2.1 and 2.2.

The authors should be clearer and more specific about the limitations of all their steps. Starting from active positioning, but also about the accuracy of all data sets published. … Data without accuracy information is no data.

The accuracies of the roll, pitch and heave measurements from the stabilization platform are reported in section 2.4. Providing accuracies for radar data is actually a research topic itself due to the fact that such variables are non-linear functions of radar raw data. Acquistapace et al. 2017 (https://doi.org/10.5194/amt-10-1783-2017) investigated the sensitivity of the Doppler moments, with a particular focus on the skewness with respect to different spectral resolutions and integration times during identical time intervals to quantify the accuracy of Doppler higher moments. Recently a paper from Myagkov and Ori (https://doi.org/10.5194/amt-2021-225) investigated how to characterize random errors in dual-polarimetric spectral observations using error covariance methods. Radar moments like mean Doppler velocity, spectral width and skewness accuracies might be characterized similarly but as shown, this is a research paper itself and goes beyond the scope of this work.

Our data paper follows the path traced by similar older radar data papers like Neto et al. 2019 (https://doi.org/10.5194/essd-11-845-2019) where no discussion on accuracy or precision of radar reflectivity is mentioned. In our work, we aim at reporting the observations that were collected, discussing all their limitations and uncertainties throughout the text. We welcome the reviewer to suggest any possible approach that they have in mind to tackle the quantification of the accuracy for the derived radar variables.

In section 3.1 and 3.2 the nomenclature should be checked again. I have the impression that not all nomenclature is used correctly and some quantities are labelled in different ways.Please check, if all appendices are mentioned in the text. I only found references to App. A, C, D.

We checked through and we found some little discrepancies in the way the reference systems are called in the text and appendices, that have been fixed. We also noticed that appendices were not referred and we added that. We could not find any additional discrepancy. We thank you for the indication.

Minor and specifics:

Minor language editing will be needed as sentences are ill-constructed from time to time.

l.6: I would prefer not to use manufacturer product names in the abstract. The "PRO" in "MRR-PRO" is not needed and it's not introduced. Stay more general and leave out the "PRO" until you introduce the product in the "Experimental setup" section.
Thank you for the comment. We removed the PRO until the instrument was introduced and then we left the MRR-PRO.

l.20: "oceanic eddies". These show up uncommented and seem to be important. Can you add a sentence on what it is and why it is interesting?
We modified the sentence as follows: "The ship sampled some mesoscale oceanic eddies, that are circular fronts of sea surface temperature anomalies caused by oceanic turbulence, locally impacting near-surface wind, cloud properties and rainfall (Frenger et al., 2013).

L22: "OA". Please introduce.
We added the sentence: "Within EUREC4A, the Ocean-atmosphere component (EUREC4A-OA, https://eurec4a.eu/overview/eurec4a-oa/) was granted two research vessels (RVs) in the Atlantic sea south-east of Barbados to monitor the oceanic processes induced by large-scale oceanic eddies."

l.28: "MRR-PRO". As before.
Corrected

L33: "… spatial and time resolution of the entire precipitation life cycle." Of what? Daily, seasonal, global, local?

Here, we refer to the fact that both radars have high temporal (1-3s) and vertical (<10 m) resolution.  Such high resolution allows to characterize and detect the processes occurring in small radar volumes, and therefore gives the ability to detect with a high sampling rate the precipitation cycle from the onset of rain until the moment in which that rain is reaching the sea surface or evaporating. The time scale of such process is hours, so it is not referred to any statistics, but more to the ability of detecting details of processes that were not discernible before.

l.42/43: "9 s". Where does this information come from? Measurement during the campaign? Does the "(Chris Fairall,…)" refer to them? Because the fact that a wavelength of 9 s needs measurements of at least 2 s to represent is mathematically obvious and would not need any support.
The 9s is precisely the information that Chris Fairall gave us from his experience in previous measurement campaigns. It refers to the longest integration time to be set in the radar for being able to correct for ship motions in the data. Although it might seem obvious, experimentally also other values like 2.5 s, or 3 s could seem acceptable. The threshold provided by Fairall was an important reference for setting up the measurement mode and define the chirp table. We reported in the publication because it might be helpful for future deployments as well.

l.47: "additional measurements onboard". Please add a "not presented/published here".
corrected

l.61-63: "Active remote … satellite retrievals." This is a bit repetitive. Please remove.
Removed

l.74-77: "We calibrated … factory calibration." This should be part of the instrument specific 2.1 and 2.2.
moved there, thanks.

l.81: "We launched …". This reads as if you will also present these for a moment. Please be precise what to expect from this manuscript.
We used the verb "we launched" because the radiosondes operation were coordinated and organized on the RV by Acquistapace, i.e. the corresponding author of the paper. Moreover, the radiosonde data are used in the paper for retrieving the IWV as stated in the next sentence and are clearly referenced in the corresponding radiosonde paper, where Acquistapace is co-author. For all these reasons, we do think that the usage that is done in the publication of the radiosonde data is clearly stated and that it would sound weird to describe such data in a different way, given the above-mentioned conditions.

l.81: "descents". What is this?
As explained in Stephan et al, 2021, during EUREC4A not only the ascents of the radiosondes were used to collect observations of P, T, RH but also the descents. By means of a parachute located inside the balloon inflated from the ground, the sonde could fall gently after the breakup of the balloon at the highest point in the atmosphere, usually between 8 and 12 km. For each radiosonde thus, theoretically, we could obtain 2 profiles of the atmosphere.

l.83: More pieces pop up. You never said that you will provide an IWV data set, did you?

We corrected that and now it is written in the introduction, together with LWP.

l.107: The ANN retrieval of the manufacturer is not documented anywhere? No reference?
The ANN retrieval is not yet published or documented by RPG.

l.114: Didn't you say that the atmosphere is relatively gas-transparent. How big are your errors? Usually other frequencies are used for IWV! Please comment. This probably only works, because of your dense and closely related radiosonde data? Which different radar "reflectivity values" are you talking about? Why plural?
The atmosphere is relatively transparent in the W-band (94 GHz), as shown in the figures below, but you can see that the response varies depending on the amount of humidity. This is why the passive channel at 89 GHz can somehow detect different IWV values in different conditions. However, much better retrievals can be obtained using a microwave radiometer and exploiting more channels between 22 and 32 and 51 to 58 GHz. Billault-Roux and Berne, 2021 (https://doi.org/10.5194/amt-2020-311 ) shows a similar approach using neural networks and also documents the lower accuracy of such approach based on a single frequency.

In our case, IWV is retrieved using a linear regression from IWV measurements from radiosoundings, collected during clear sky. To determine whether there are clouds or not, we use the Wband radar reflectivity, that is very sensitive to small cloud droplets. For a given time stamp to be considered clear-sky, all radar reflectivity observations collected in the vertical column of atmosphere should be < -50 dBZ. If such condition is met, then the time is classified as clear sky, and the radiosonde launched at that time is considered in the dataset for retrieving IWV.

[Figure]

sources for the figures: left : https://bit.ly/3qQV6te , right: https://bit.ly/3kTfVQV

l.132: I'm missing the products IWV and LWP.
LWP is mentioned among the integrated variables while the IWV is not stored in the output file because it has been derived just for the paper. The variable stored for future application is the brightness temperature at 89 GHz, as listed among the integrated variables.

l.142: Only this is where you should introduce the "PRO" part in MRR-PRO. "PRO" most likely is the "Professional" one. Only manufacturer terminology, but tell us.
As explained in the manufacturer page relative to the instrument, https://metek.de/product/mrr-pro/ the PRO stands to indicate that the new version of the

MRR is the development of the old MRR with the technique of the MRR-2 that displayed higher performances and significantly improved parameter estimations. After introducing the PRO suffix, we prefer to keep it in the rest of the paper, to highligh that the instrument deployed belonged to the new generation of MRR instruments, with much higher capabilities.

l.146: This sentence seems awkward. Please re-phrase.

At line 146 we found this sentence: "Details on the ship motion correction algorithm and the interference filter are provided in section 3.2 and 3.4, respectively. The data are organized in daily files and the variables provided after the processing chain described in Figure 3 are reflectivity considering only liquid drops, equivalent reflectivity non-attenuated, equivalent reflectivity attenuated, hydrometeor fall speed, spectral width, skewness and kurtosis of the Doppler spectra, liquid water content, rainfall rate, rain drop size distribution, raindrop diameter weighted over mean mass, time, height, latitude and longitude.

We removed Figure 3 and explained the postprocessing more extensively in the text, followed by the list of variables.

Fig. 3: This is "method" not "experimental setup", isn't it? Wouldn't this better fit into 3.2 or 3.4?

We changed the title of the section to "experimental setup and data processing", thank you for your comment.

Tab.1: "spectral bins", "spectral resolution". This refers to the Doppler Fourier spectrum?

The number of spectral bins is the number of points in the fft transform to derive the Doppler spectrum. They correspond to the resolution of the xaxis along which the Doppler spectrum is displayed. The spectral resolution, in ms$^{-1}$, expresses what is the width in ms$^{-1}$ of a single spectral bin. The ensemble of spectral bins composes the x axis along which the Doppler spectrum is displayed.

Please clarify.l.167: At the end of this section. What are the limitations of this positioning correction?

The limitations of the stabilization platform correction are stated in the sentence: "It must be noted that the stabilization platform can compensate for the rotation of the ship but it cannot compensate for the vertical movements along the vertical axis (heave, etc.) and the translations which occur because the ship rotates around its center of mass while the instruments are located elsewhere (see Section 3)"

In the rest of the paper, we discuss extensively the limitations of the table's ability to correct also for the rotation of the ship. The paper is almost entirely about that.

What is the frequency of correction steps? And how does this fit to expected and observed roll and pitch values and radar sampling? Where do the mentioned 35% come from? Please clarify.

Correction is applied to each chirp radar time stamp. This means that is it applied at a time resolution higher than the time resolution of the radar, that is 3s. Specifically, the chirp integration times are given in Table 3. Let's consider a given time stamp of the radar from T1 to T2. The corrections are applied:

- at T1+0.846 for the chirp1 (radar range gates between 100 and 1233 m),
- at T1+0.846+0.786 for the chirp 2 (radar range gates between 1233 and 3000 m)
- at T1+0.846+0.786+1.124 for chirp 3 (radar range gates between 3000 and 10000m).

Note that the sum of the chirp integration times is 2.756 s, less than 3 s. The difference is an additional buffer time internal to the radar.

Roll and pitch values are provided with 1s resolution and the ones assigned for calculating the correction are the closest to the correction time stamp, as defined above, dependent on the chirp. It is clearly evident, thus, that using a higher resolution for the ship data (0.1 s instead of 1s) is crucial for improving the quality of the correction.

The 35% amount comes from the sensors of the table. Table measures roll and pitch as a function of time and whenever the table stops, no data are collected. By looking at the stable table time serie of data, one can easily calculate the amount of time the table stopped and the amount of time it was working.

Top paragraph on page 9 (line numbering mixed up): What is the frequency of MRU position measurements? You have not mentioned it anywhere.
It is 1s, it has been added. Thanks for the comment.

Page 9: There seems to be an 1,70 m offset between W-band and MRR. The image does not show this. Please comment.
Thanks for noticing the mistake. We assigned the same height to the two instruments, that is -17.40. The other measurement was taken when the stable table was off. The hydraulic pump that controls the table pistons, when turned on, pushes up the table and the instruments.

Fig 5: The combination of axes sketch and image in the back is confusing. First it looks as if you show a pitch angle. Then it takes some time to detect that you show an arbitrary combination of all three. Can you improve that? Maybe by changing the ship display and/or moving it away from the center of the figure? And where is the arrowhead of the y-axis?
Thank you for the comment. We tried in the new version of the manuscript to improve the figure. The arrowhead of the y axis, due to the 3d perspective, is not visible because it is in the dark circle. We modified the image.

l.200: "downdraft". The radar rather sees the downward motion of droplets and not the downdraft of the air. Please correct.
Cloud droplets have negligible fall velocities and therefore can be considered as air tracers. It cannot exist a downward motion of cloud droplets without a corresponding downward air flux. This is why generally we can talk about downdrafts also using radar observations. The measured fall speed is always a convolution of the air speed and the droplet fall speed. When drops have a non-negligible fall speed the radar measured velocity is a convolution of their fall speed and of the air velocity. Since rain while falling advects also air, there is also in this case a downdraft associated with the precipitation. This is why we talk of downdraft. We believe that this is the term that better represents the physical processes occurring in the situations observed in this paper.

L.215: Whole paragraph. Above you stated that you use a 20 min time window. Here you state that it is done for every radar chirp. Please explain at what frequency you derived the DeltaT. And please, say a few words about likely reasons of the time offset. I can imagine that a time offset of several seconds between instruments develops over time. An erratic variation over the campaign is something I would not understand.

We thank the reviewer for the comment. The correction is applied to hourly files. For each hour, we estimate what is the time gap that every chirp has (chirp time stamps are clarified above). Such time gaps are different, and this is why we have to do it for each chirp. We estimate one deltaT for each chirp, so in total, 3 DeltaT for every hour. To calculate a DeltaT, we consider a 10 minutes (sorry for the mistake, not 20) time serie of velocity measurements. The 10 minutes duration is necessary to guarantee enough data for the variance calculation, it corresponds to 200 time stamps (at 3 sec resolution). This choice is also due to the nature of these clouds, that often do not last more than 20 minutes in the observations. We average the time series over the chirp range gates, as described in the text, and calculate the variance. The DeltaT minimizing the variance var(DeltaV) is the delay for the chirp. The procedure is repeated for each chirp. Example time series are represented in Figure 6.

Regarding the reasons for the offset: We did all what was feasible to do on the ship to synchronize the instruments. The offset was observed and it is also visible in Figure 6 when comparing <vd> and W_signal. We hypothesize that it might be due to the hardware connections and computer synchronization, but we cannot prove this. Also, as mentioned in the chirp time description, there is an additional 0.3 s time that is necessary for the radar to switch from one chirp to the other and to a new chirp cycle. Some variability can be due also to this internal radar processing time.

l.233/234: Didn't you just explain the same facts using the terms v_d and v_hyd in section 3.1? Please explain the difference once more, if needed. Otherwise please adjust nomenclature and remove repetitions.
We did not really understand to which facts the reviewer is referring to. In line 233-234 there is "The Doppler velocity measured by the radar is the projection of the particle's velocity vector on the radar line of sight. Therefore, the component of the velocity vector of the hydrometeors w signal measured by the radar is positive when hydrometeors move upwards. "We thought that it would have been good to state again this simple convention before entering in a long derivation of vectorial equations where the sign of the vertical axis is actually crucial. Is this the point the reviewer addresses and the sentence to remove?

The reason why the concept is repeated is the following. The complete vectorial equation for the velocity observed by the radar is equation 2. For calculating the time shift, the formula is simplified by neglecting all v_radar components other than v_heave. This approximation is valid since the other terms V_course and V_rot are much smaller than v_heave.
To clarify better this point, we added the following sentence in section 3.1 from line 200 in the new paper version :
"By comparing the heave rate (thin blue line) and< Vd> time series (thick red line in Figure 6) we can derive the time lag ΔT. Cloud droplets have a vertical speed of w_hyd. The ship

is moving vertically due to waves with w_heave. To a first approximation, we can neglect additional contributions to the vertical motion of the ship (for the full vectorial equation see treatment in section 3.2). The radar measures Doppler velocity vd with respect to the instrument on the ship, hence vd = whyd + wheave.

We hope to have clarified the raised point. If not, we welcome suggestions from the reviewer.

l.259: Please comment on the need to label this data as "limited quality" here.
We thank the reviewer for the comment. In this part we just described the approach and we did not enter in the discussion of the quality of the correction. We just present how the correction is performed. Of course, such data have limited quality. However, this point is touched in the discussion on the plot of Figure 7, where the limitations of the approach when the table is not working are clearly displayed. This is why we did not opt for expanding on the data quality at line 259.

Page 13, bottom (line numbering broken again). "intensity" à "magnitude" of a vector?
We substituted intensity with magnitude.

L.268, eq 8: Reads like vectors are subtracted from a scalar?! It should be italic "v"s in the equation 8. On the other hand, it is not intuitive to label the z-component with another letter that "w"? Do you need another definition here? Please clarify and simplify nomenclature.
W is the z component. w_signal is coherently defined along z, while the other two terms are now in italics, representing the z components of the corresponding vectors. We believe the notation in this way is clear. We are sorry for the typo in the equation 8.

L.280: The first part of 3.3 reads slightly repetitive, apart from the horizontal wind influence. I think this part would better fit as last summarizing point in section 3.2.
l.284-292: This is why I'm still missing some information on accuracy of all these corrections in the different situations and some suggestions how to deal with them when using the data! Please add this information.
We decided to include the first part of 3.3 there as a sort of example application of the formulas derived in section 3.2. We believe it is better to separate the parts because section 3.2 is quite long already. We also think that it is important to discuss extensively the cases when the table is working and not working, supporting the discussion with Figure 7.

Regarding the accuracies: it would be fantastic to be able to provide an accuracy, because it would mean that we know the truth, regarding the corrections. Unfortunately, it is not so, and there is no reference to which we can compare to evaluate the correctness of the correction terms. We welcome any suggestion that can come from the reviewer on how to quantify the accuracy of the correction. From our side, we did our best to be rigorous and get the best data in the given conditions.

l.296: "Figure 7d)-h)." No d) and h) only show the platform status. e,f,g show the performance.
Now corrected to Figure 7 e)-g). We thank you for the comment.

l.302: "as described by looking at..." and Fig 8. .... I'm confused. I assume that this is a Fourier spectrum of the time series of vertical motion? Correct? In a certain cloudy range gate you can see the Doppler velocity fluctuation combined from particles' fall speed and ship motion w_signal. You cannot see the "vertical motion of the radar" v_rad which results from the ship motion, can you? Please mention equations 2, 7 and 8 and explain better.Fig. 8: I guess it should be frequency in "1/s" or better "Hz" on the x-axis? And the top
label "periods" à "period".

The figure shows the fast Fourier transform of various quantities. In panel a) the black line is precisely the FFT of the nominator of the correction term in equation 6, i.e. (V_wind_s – V_radar) * ep. Generally, V_wind_s has no vertical component (see appendix B), so the scalar product with Ep, when the table works, as it is the case for the figure 8, results in the z component of the vector v_radar, called v_radar_z.
v_radar_z is the resulting velocity "felt" by the radar, and it is given by the composition of the various contributions along z of the terms given in equation 7, namely V_trans = [0,0, w_heave] and V_rot = [v_rot_x, v_rot_y, v_rot_z]. Such contributions are w_heave and v_rot_z.

We compare the FFT of the v_radar_z with the FFT of v_rot_z and w_heave, to demonstrate that the main contribution to the signal comes from w_heave. The FFT of the V_radar_z (black) is basically almost entirely overimposed to the one of the heave (translational velocity, purple line). It means that the heave rate is the main contribution to the vertical motion of the radar, that is on the ship.

In panel b) we compare the signal measured by the radar w_signal (black line here, while in panel a) it was represented by the yellow line) with the FFT of the velocity seen by the radar after applying the correction with the time shift v_hydr, while in panel c) it is the same thing but without the time shift correction. The comparison of panel b) and c) is presented to show the impact of the time shift correction. Basically, the frequencies due to the waves are not removed in panel c, while in panel b they are smoothed in the pink line.

We changed the labels in the figure according to the explanation above and we modified the caption in the new pdf version. We thank the reviewer for the comment that helped us to improve the description of the figure. We hope that now it can be easier to understand the showed results.

l.308: Is this the right range? 0.5 Hz is in the middle of your x axis!? See next point.
0.2 Hz lays on the right edge of the lower axis, in correspondence of 5 s period. The 0.5 Hz is not included in the range of displayed values.

l.310: This smooths the part between above 0.1 = 10 s period, correct? Above you state the range 0.1 to 0.5 Hz which would be period range 10 s - 16 s !?
I am not sure I am getting right the reviewer's comment here. The range 0.1 to 0.5 Hz corresponds to the range between 10s and 2s. We display the range between 10 and 5 s, which shows the mentioned increase of the spectra towards the Nquist frequency.

l.311: Confusing sentence. And basically, nothing you need to put into an equation. Mean

speed multiplied with time is distance. Remove it.

We thank the reviewer for this comment. We tried our best to discuss the limitations of the presented approach and quantify them. The horizontal resolution for the radar is a distance, and therefore this is why it was expressed as a product of velocity times the integration time.

We reformulated removing the product and the sentence reads now as follows: "However, the 9 s smoothing degrades the average horizontal resolution of the $V_{hyd, mean}$ by a factor of 9. For an average ship speed of 3 $ms^{-1}$, the resolution would change from 3 m to 27 m, resulting in a slightly higher resolution than the vertical 30 m one. However, daily maximum speeds for the ship can reach also 9 $ms^{-1}$, producing thus a coarser resolution."

l.333: "Prominence" is not a mathematical term. Please define.

Prominence is defined exactly after mentioning the word, as the peak's ability to stand out from the surrounding baseline of the signal. This is the most understandable definition, that is also referenced in the description of the python routines to calculate such quantity: https://docs.scipy.org/doc/scipy/reference/generated/scipy.signal.peak_prominences.html

L339: "lowest 600 m". Please give the reason why this assumption/ condition is justified.

The reason for asking for continuity in the lowest 600 m for the mean Doppler velocity of the MRR is that this it approximately the height of the sub-cloud layer, where rain gets out of the cloud and reaches the ground/evaporates. MRR detects only rain, and had strong interference problems above such height. This is the range of heights where we are sure that the MRR can detect rain, so it is worth comparing in this region.

l.342: "abrupt"? Please explain a bit. What reason could there be to accept 2 or 3? The same is true for point 3 in l.343. At least explain why, if they are all found empirically.

All these conditions are posed empirically to remove the signatures due to interference. The way in which the interference pattern affects the mean Doppler velocity of the MRR is displayed in figure figure A1). In figure A1a) the mean Doppler velocity field is continuous where there is signal, and is it a sequence of very high and low values where there is noise. If one calculates the difference of consecutive values along the profile, it is clear that peaks appear whenever nearby values change from extremely high to low or viceversa. When there are just a few peaks (3, or 4) , it might still be the case that there is a real signal in the column. However, when there are more than 8, we experimentally found that those columns correspond to interference (like in Figure A1a) at 1:50.

l.373: I wonder if "flower" has made it to general technical language already. Please explain EUREC4A slang.

We added a peer-reviewed paper published where the flower type of cloud is introduced.

L378: "difficult to quantify". Uuh. Isn't this what a data publication should provide? Best estimates of accuracy of the published data? If you cannot provide any accuracy you should not publish it at all? Please think about a way to provide some estimate on this.

We already answered this comment at the very beginning, stating that providing an accuracy to radar moments is itself a research topic out of the goal of this publication. We also provide information on the correction and its limitation, for what is feasibly possible, as describe in the comments above.

l.385: "Fig 12 b) displays clear areas". I'm confused. I don't see clear areas in b), I see them in a)? In addition, some explanation would be nice. Why is it clear in a) when I see large negative values in b)?
Thank you for the comment. We realized of an error in the plotting. We added and commented a new figure.

l.399: Please explain. LWP includes the rain. How does it contaminate the measurement?
Rain can make the radome of the 89 GHZ channel wet, degrading the signal. Normally values above 1000 gm-2 are flagged as rain.